

# Class-weighted Dempster–Shafer in dual-level fusion for multimodal fake real estate listings detection

Maifuza Mohd Amin, Nor Samsiah Sani and Mohammad Faidzul Nasrudin

Science and Information Technology, Universiti Kebangsaan Malaysia, Bangi, Selangor, Malaysia

## ABSTRACT

**Background:** Detecting fake multimodal property listings is a significant challenge in online real estate platforms due to the increasing sophistication of fraudulent activities. The existing multimodal data fusion methods have several limitations and strengths in identifying fraudulent listings. Single-level fusion models whether at the feature, decision, or intermediate level struggle with balancing the contributions of different modalities leading to suboptimal decision-making. To address these problems, a dual-level fusion from multimodal for fake real estate listings detection is proposed. The dual-level fusion allows the integration of detailed features from text and image data to be performed at an early stage, followed by the metadata fusion at the decision stage in order to obtain a more comprehensive final classification. Furthermore, a new weighting scheme is introduced to optimize Dempster–Shafer in decision fusion to help the model achieve optimal performance and as a result, our method improves the classification. The Dempster–Shafer without class weightage lacks the flexibility to adapt to varying levels of uncertainty or importance across different classes.

**Methods:** In Class Weighted Dempster–Shafer in Dual Level Fusion (CWDS-DLF), we employ advanced models (XLNet for text and ResNet101 for images) for feature extraction and use the Dempster–Shafer theory for decision fusion. A new weighting scheme, based on Bayesian optimization, was used to assign optimal weights to the 'fake' and 'not fake' classes, thereby enhancing the Dempster–Shafer theory in the decision fusion process.

**Results:** The CWDS-DLF was evaluated on the property listing website dataset and achieved an F1 score of 96% and an accuracy of 93%. A t-test confirms the significance of these improvements ($p < 0.05$), demonstrating the effectiveness of our method in detecting fake property listings. Compared to other models, including 2D-convolutional neural network (CNN), XGBoost, and various multimodal approaches, our model consistently outperforms in precision, recall, and F1-score. This underscores the potential of integrating multimodal analysis with sophisticated fusion techniques to enhance the detection of fake property listings, ultimately improving consumer protection and operational efficiency in online real estate platforms.

Corresponding authors
Maifuza Mohd Amin,
p123261@siswa.ukm.edu.my
Nor Samsiah Sani,
norsamsiahsani@ukm.edu.my

## INTRODUCTION

In the digital age, where online platforms serve as the primary marketplace for property transactions, the proliferation of fake property listings has emerged as a significant challenge. As seen in prominent platforms like eBay and Craigslist, instances of counterfeit real estate advertisements have surfaced (*Baby & Shilpa, 2021*). These deceitful practices often involve manipulating data by presenting high-end properties at significantly reduced prices and in desirable locations, capturing the interest of budget-conscious buyers seeking value for their money. These deceptive listings not only mislead prospective buyers and renters but also undermine the integrity and trustworthiness of the real estate market (*Ringkardo, 2023*). The Internet Crime Complaint Centre of the Federal Bureau of Investigation reported that in 2021, over 11,578 individuals fell victim to real estate fraud, resulting in losses totaling $350 million (*Internet Crime Complaint Center, 2021*). The misuse of real estate images for fraudulent transactions has also been reported (*Magdelin, 2022*). To combat this pervasive issue, there is an urgent need for advanced detection mechanisms capable of effectively identifying fraudulent listings that have been intermingled with legitimate ones. Fraudulent listings refer to property advertisements designed to deceive potential buyers or renters by providing false or misleading information about a property or its ownership (*Mohd Amin et al., 2024*). Examples include listings for properties that do not exist or are misrepresented.

Various studies have been conducted to detect fraudulent activities using multimodal data. However, in the real estate domain, most counterfeit prevention efforts have focused on developing secure real estate transaction environments using blockchain technology (*Joshi & Choudhury, 2022*; *Vivekrabinson et al., 2023*; *Panwar et al., 2024*; *Shehu, Pinto & Correia, 2022*). Nonetheless, there is a growing interest within the research community to develop systems for detecting fraudulent real estate properties using machine learning and deep learning methods, encompassing data mining, fake image manipulation detection, and real estate data analysis (*Rutzen, 2023*; *Smart Realty, 2023*).

Studies on detecting real estate fraud using machine learning have focused on unimodal data, specifically metadata as the primary modality (*Mohd Hamim & Sani, 2022*; *Nguyen-Duc, Nguyen & Nguyen, 2023*). Due to the limitations of these existing studies, this research adapts approaches from fake news detection in social media (such as Twitter, Instagram, Facebook, and others), as this domain is more robust and has been extensively refined by numerous researchers. Enhancements in this study are made by refining relevant approaches to the unique characteristics of real estate data, which differ significantly from social media data.

Furthermore, the existing multimodal fusion techniques at the feature, decision or intermediate level struggle with balancing the contributions of different modalities leading to suboptimal decision making (*Bodaghi, Hosseini & Gottumukkala, 2024*; *Boulahia et al., 2021*). Features fusion involves combining raw input data or features at an early stage. While this approach can capture correlations between different modalities, it often leads to high-dimensional feature spaces that are computationally expensive and prone to overfitting (*Guarrasi et al., 2024*). Additionally, it may not effectively handle situations

where different modalities contribute unevenly to the final decision, potentially diluting important information (*Liu et al., 2024*). While decision fusion involves making independent decisions for each modality and then combining these decisions at a later stage. This method is simpler and more modular, but it can miss important interactions between modalities and might not fully exploit the complementary nature of the data sources (*Zhang et al., 2024*). Meanwhile, the mid-level fusion tries to balance between early and late fusion by combining features at a mid-stage. While this approach aims to capture the strengths of both early and late fusion, it can be difficult to determine the optimal point of fusion, and it might still suffer from issues like overfitting or suboptimal use of modality-specific information (*Guo & Song, 2022*).

In the pursuit of enhancing the accuracy of fake real estate listing detection, a dual-level fusion model using multimodal data is proposed. This model addresses the common limitations of early, late, and mid-level fusion approaches by integrating the strengths of both feature and decision-level fusion. Such integration allows for a more refined and effective combination of multimodal data, making the model more robust and adaptable across various tasks (*Mohd Amin et al., 2024*). Additionally, a new weighting scheme is introduced to optimize the Dempster–Shafer technique in decision fusion, further improving the classification performance. These advancements are critical in developing a more reliable and efficient detection system. The contribution of this study can be summarized as follows:

- This study lies in its innovative approach to combining multimodal information. The Class Weighted Dempster–Shafer in Dual Level Fusion (CWDS-DLF) employs two levels of combination within its model: at the feature level and the decision level. Initially, the integration of text and image data occurs at the feature level. Subsequently, the output from this feature combination is further merged with property metadata at the decision stage. This comprehensive approach enables a thorough evaluation of genuine and fake listings. Moreover, the CWDS-DLF introduces the weights with the Dempster–Shafer combination algorithm to address the imbalance class in dataset. Consequently, the model's performance is significantly enhanced.
- CWDS-DLF was implemented using real estate datasets provided by Durian Property, a real estate industry company that lists properties from various states across Malaysia. This demonstrates that CWDS-DLF can assist the real estate industry in detecting fraudulent property listings more effectively.
- The performance superiority of CWDS-DLF is tested and compared with other multimodal and unimodal data analysis models.

To provide readers with a clear guide through this research, the article is structured into the following main sections: "Background and Related Work" is a literature review of recent works on fraud detection across various domains. "Methods" the methodology of the study. "Experiment Setup", a detailed description of the experiment setup. Finally, the "Result Discussion" and "Conclusion" are presented.

## BACKGROUND AND RELATED WORK

In recent years, multimodal data analysis studies have become prevalent across various fields including healthcare, medical science, e-commerce, engineering, and others (*Lahat, Adali & Jutten, 2015*). Multimodal data analysis involves integrating multiple sources of information to glean deeper insights and enhance decision-making processes (*Castanedo, 2013*; *Freedman, 1994*; *Gudavalli et al., 2012*; *Jusoh & Almajali, 2020*; *Liu et al., 2022*; *M'Sabah, Bouziane & Ferdi, 2021*; *Roitberg et al., 2022*; *Zhu et al., 2020*). Encouraged by these successes, numerous studies on fake detection have been conducted across various domains (*Athira et al., 2022*; *Duc Tuan & Quang Nhat Minh, 2021*; *Fang et al., 2019*; *Jin et al., 2017*; *Singh & Sharma, 2022*; *Song et al., 2021*; *Wang et al., 2023*; *Ying et al., 2021*). This is because perpetrators often exploit multiple data modalities to deceive their victims.

### Early fusion (features fusion)

The multimodal studies outlined in Table 1 are classified into three categories: early fusion, late fusion, and a combination of both early and late fusion. Several studies employ various early combination techniques, including concatenation and attention mechanisms to detect fake news in media social (*Duc Tuan & Quang Nhat Minh, 2021*; *Li et al., 2022*; *Liang, 2023*; *Liu et al., 2023*; *Zhou et al., 2022*). The concatenate technique refers to a method of feature fusion in which features from different sources or modalities are combined by simply concatenating them together. This means that the features are joined end-to-end to create a single, longer feature vector. This technique is utilized in the study (*Agarwal et al., 2019*; *Alonso-Bartolome & Segura-Bedmar, 2021*; *Athira et al., 2022*; *Wang et al., 2023*; *Ying et al., 2021*), wherein multimodal data, such as text and images, are combined to enable the model to make more informed and accurate predictions or classifications. Before concatenation, most of these data sources are embedded using learning models such as XLNet, BERT, VGG, ResNet, and CNN. This process captures meaningful representations of the input data. These representations enable better generalization, transfer learning, and downstream performance in various machine learning tasks. In this study (*Ying et al., 2021*), for instance, text data is embedded using the Bert model, while image data is using ResNet101. Consequently, employing this approach led to a model performance with an accuracy of 90.57% on the Fakeedit dataset (https://github.com/entitize/Fakeddit?tab=readme-ov-file).

Alternatively, other early fusion method involves utilizing the attention mechanism technique. The attention mechanism in machine learning is a computational technique used primarily in the context of sequence-to-sequence models, such as cross model attention, multimodal transformer, scaled dot product attention and modality wise attention (*Aziz, Yaakub & Bakar, 2024*; *Duc Tuan & Quang Nhat Minh, 2021*; *Liang, 2023*; *Liu et al., 2023*; *Ying et al., 2021*; *Zhou et al., 2022*). It enables the model to focus on specific parts of the input sequence when making predictions or generating outputs. The study (*Liang, 2023*) employed the Cross-Modal Attention technique to fuse text and image content, achieving an accuracy of 88.9% on the Weibo dataset. However, employing the Modality-Wise Attention Mechanism technique on the same dataset can enhance the model's performance, achieving up to 90.7% accuracy (*Zhou et al., 2022*). This illustrates

**Table 1 The multimodal studies in various domains.**

| Year | Dataset | Fusion level | Model/Fusion technique | Result |
|---|---|---|---|---|
| *Athira et al. (2022)* | • Gossipcop (8,000 data)<br>• Politifact (600) | Early fusion/ features fusion | Early fusion (Concatenate)<br>• XLNET (Text embedding)<br>• VGG19 (Image embedding) | Accuracy:<br>• 81.8% GossipCop<br>• 87.1% Politifact |
| *Liang (2023)* | • Gossipcop (fake-2,549, real-10,206)<br>• Weibo (fake-4,121, real-1,054) | | Early fusion (Cross-Modal Attention)<br>• XLNET (Text embedding)<br>• VGG19 (Image embedding)<br>• BLIP-based multimodal feature extractor is employed to acquire a comprehensive multimodal feature representation from news content. | Accuracy:<br>• 88.9% Weibo<br>• 87.3% GossipCop |
| *Liu et al. (2023)* | • Fakeddit<br>• Weibo | | Early fusion (Multimodal Transformer)<br>• Pre-trained Faster RCNN and ResNet (image embedding)<br>• BERT (texts and image caption) | Accuracy:<br>• 90.57% Fakeddit<br>• 88.68% Weibo |
| *Agarwal et al. (2019)* | • Twitter (2000)<br>• Weibo | | Early fusion (Concatenate)<br>• Bert (Text embedding)<br>• VGG19 (Image embedding) | Accuracy:<br>• 77.77% Twitter<br>• 89.23% Weibo |
| *Duc Tuan & Quang Nhat Minh (2021)* | • Twitter | | Early fusion (Scaled Dot-Product Attention)<br>• Bert (Text)<br>• VGG19 (Image) | Accuracy:<br>• 81.2% Twitter |
| *Alonso-Bartolome & Segura-Bedmar (2021)* | • Fakeddit | | Early fusion (Concatenate)<br>• CNN (Text embedding)<br>• CNN (Image embedding) | Accuracy:<br>• 87% Fakeddit |
| *Zhou et al. (2022)* | • Weibo<br>• GossipCop<br>• Politifact | | Early fusion (Modality-Wise Attention Mechanism)<br>• ResNet (Image)<br>• BERT (Text)<br>• CLIP (both text and image) | Accuracy:<br>• 90.7% Weibo<br>• 88.0% GossipCop<br>• 94.2% Politifact |
| *Ying et al. (2021)* | • Weibo (fake-4,749, real-4,779)<br>• Pheme (fake-1,972, real-3,830) | | Early fusion (concatenate)<br>• ResNet (Image)<br>• BERT (Text)<br>• Cross attention network | Accuracy:<br>• 87.9% Weibo<br>• 87.2% Pheme |
| *Wang et al. (2023)* | • Twitter (fake-5,007, real-840)<br>• Weibo (fake-1,000, real-996) | | Early fusion (Concatenate)<br>• Cross-modal Contrastive Learning<br>• Cross-modal Fusion<br>• Cross-modal Aggregation | Accuracy:<br>• 90% Twitter<br>• 92.3% Weibo |
| *Nguyen-Duc, Nguyen & Nguyen (2023)* | • Text (extract metadata from text modality) | | Automated machine learning combined with two-layer stack ensemble techniques | Accuracy: 91.5% |

(Continued)

| Year | Dataset | Fusion level | Model/Fusion technique | Result |
|---|---|---|---|---|
| *Gumaei et al. (2022)* | • Patient health information, travel demographics and geographic | Late fusion/ decision fusion | Late fusion (Soft Voting)<br>• Random Forest<br>• Gradient Boosting<br>• Extreme Gradient Boosting | Accuracy:<br>• 97.24%<br>F1-score:<br>• 97% |
| *Ilhan, Serbes & Aydin (2022)* | • 3 dataset of X-ray images | | Late fusion (Majority Voting)<br>MobileNetV2,<br>• VGG16,<br>• ResNet50,<br>• ResNet101,<br>• NasNet,<br>• InceptionV3<br>• Xception | Accuracy:<br>• Dataset 1 - 90.8%<br>• Dataset 2 - 90.5%<br>• Dataset 3 - 90.7% |
| *Oh & Kang (2017)* | • 3D point clouds and images | | Late fusion (Basic Belief Assignment)<br>• CNN | Average precision:<br>• 77.72% |
| *Chen et al. (2022)* | • Audio dan video | | Late fusion (LSTM)<br>One-dimensional CNN-audio<br>Two-dimensional CNN-video | Accuracy:<br>• 77.07%<br>F1-score: 75.6% |
| *Rwigema, Mfitumukiza & Tae-Yong (2021)* | • Image | | Late fusion (Majority voting, Naïve-Bayes combination and Sum rule)<br>• ANN-gender<br>• CNN-age | Accuracy:<br>• Majority voting - 81.2%<br>• Naïve Bayes - 85.5%<br>• Sum-rule - 86.1% |
| *Luo et al. (2022)* | • Image (head position and face expression)<br>• Interactive data | Combination of early and late fusion | • Hierarchical Random Forest—Head movement<br>• Conditional Random Forest—feature emotions<br>• Weighted Hierarchical Fusion—a technique of combining thought features and both decision combinations | Accuracy:<br>• 87.5% |
| *Yala et al. (2019)* | • Mammogram images<br>• Patient's medical records | | • Risk Factor Logistic Regression (RF-LR) - risk factor<br>• Convolutional Neural Network (Resnet18) with Pytorch—mammogram images<br>• Feed Forward Neural Network—a hybrid combination of traditional risk factors and mammograms | Accuracy:<br>• 70% |
| *Du et al. (2019)* | • Highways England dataset (Location, date, time period, speed, flow and travel time and so on) | | • CNN-GRU-Attention;<br>Traffic flow, speed and travel time<br>• Combined model;<br>Adaptively Joint Model—uses mathematical techniques as a joint model. | Root means square error:<br>• 4.35 |

| Year | Dataset | Fusion level | Model/Fusion technique | Result |
|---|---|---|---|---|
| **Table 1** (continued) | | | | |
| Proposed study | • Durian property (image, texts and medata) | | • XGBoost-metadata<br>• XLNet-text<br>• ResNet-image<br>• Features fusion-ANN<br>• Decision fusion-Dempster–Shafer with weight | Accuracy:<br>• 93%<br>F1-score:<br>• 96% |
| *Mohd Hamim & Sani (2022)* | • Multi data set (EdgeProp and DurianProperty real estate listings with 124,735 records) | Unimodal analysis | • K-min clustering, Euclidean Distance calculation<br>• Support Vector Machines | Accuracy:<br>99.73% |

that the effectiveness of the attention mechanism technique depends on how well it aligns with the specific characteristics of the dataset to achieve optimal classification performance. Moreover, some studies utilized specific techniques to generate additional modalities. For instance, Bootstrapping Language-Image Pretraining (BLIP) is employed to acquire multimodal feature representation in news content (*Li et al., 2022*). Similarly, another study (*Zhou et al., 2022*) utilized Contrastive Language-Image Pretraining (CLIP) to generate multimodal features.

## Late fusion (decision-level fusion)

Some studies utilized combined methods at the late fusion (decision level). Decision-level fusion techniques, such as majority voting, determined the final results by aggregating outputs from each source or classifier through majority voting. The decision with the highest number of votes is selected as the final decision. The studies by *Ilhan, Serbes & Aydin (2022)* and *Rwigema, Mfitumukiza & Tae-Yong (2021)* utilized this technique to combine results from several classifiers, including MobileNetV2, VGG16, ResNet50, ResNet101, NasNet, InceptionV3, Xception, and ANN. The model's performance is found to be optimal in the study by *Ilhan, Serbes & Aydin (2022)*.

The weighted voting technique, like majority voting, assigns a weight to each decision from every source based on its reliability or accuracy (*Ding et al., 2022*). The results are then combined by multiplying each decision by its corresponding weight and summing them. The decision with the highest total weight is chosen. This technique was utilized by *Gumaei et al. (2022)* in their study, wherein they combined classifiers such as random forest (RF), gradient boosting (GB), and Extreme Gradient Boosting (XGB). The Dempster–Shafer theory offers a mathematical framework for managing uncertainty and conflict in decision-making. It employs a confidence function to depict the level of confidence in different hypotheses and integrates them based on the evidence provided by various sources. The study by *Oh & Kang (2017)* applies this theory by incorporating the basic belief assignment (BBA) coefficient in the calculation.

Bayesian techniques use probability theory to combine results or predictions from different sources (*Kim et al., 2022*). The results are combined by calculating the probability distribution and the observed data using Bayes theorem. The studies by *Rwigema, Mfitumukiza & Tae-Yong (2021)* also employ this technique alongside other combined techniques, such as ensemble methods and majority voting. Neural networks can facilitate decision-making by training a model to discern the relationship between inputs from various sources and the desired output. Such networks can make decisions based on the combined information. Similar to the study by *Tang, Xu & Chen (2021)*, which employs LSTM to integrate two classifier models: one-dimensional CNN for audio and two-dimensional CNN for video.

Ensemble techniques like bagging, boosting, and stacking combine results from multiple models or classifiers to enhance overall performance (*Shah, Patil & Dongardive, 2024*). Each model generates results, and the ensemble aggregates them using various techniques. A study by *Rwigema, Mfitumukiza & Tae-Yong (2021)* concluded that this fusion method is more effective than majority voting and naïve-Bayes decision fusion. All the mentioned decision fusion techniques employ distinct approaches to combine information from diverse sources. It is important to note that each study utilizes a unique combination of these techniques. The selection of techniques is contingent upon factors such as the problem's characteristics, the nature of the data or results, and the available information sources. Experimentation and evaluation are necessary to ascertain the most suitable technique for a specific application.

## Early and late fusion integration

By employing a combined method that integrates both early and late combination techniques, the advantages of both approaches can be realized. This method has been utilized in several studies, including those by *Du et al. (2019)* and *Yala et al. (2019)*. In a study by *Yala et al. (2019)*, a CNN model was employed to learn image features, which were then combined with clinical features before being fed into a Feed Forward Neural Network model. The combination of images and clinical features is accomplished using the Simple-Vote technique. In a study by *Luo et al. (2022)*, video format data sources and student evaluation score values are utilized to predict student interest in class. Head movement in the video serves as a measure of student attention, while smile recognition is used to capture emotions. All of these video data sources were analyzed using hierarchical random forest for head movement and conditional random forest for smile recognition learning models. The results from the learning models' analysis are then combined using the Analytic Hierarchy Process (AHP) technique. The cognitive feature is derived by initially combining the score feature and the accuracy (time) of a student's assessment with an aggregation technique using weights. Finally, the combination of attention, emotions, and cognitive features was integrated using the AHP technique to determine the study's outcomes. An accuracy of 87.5% was achieved. In contrast to the study by *Du et al. (2019)* which utilized the mathematical technique known as the Adaptively Joint Model, a root mean square error value of 4.35 was obtained. Overall, it was observed that the model's performance improved. The diversity of modality and analysis models can be enhanced by
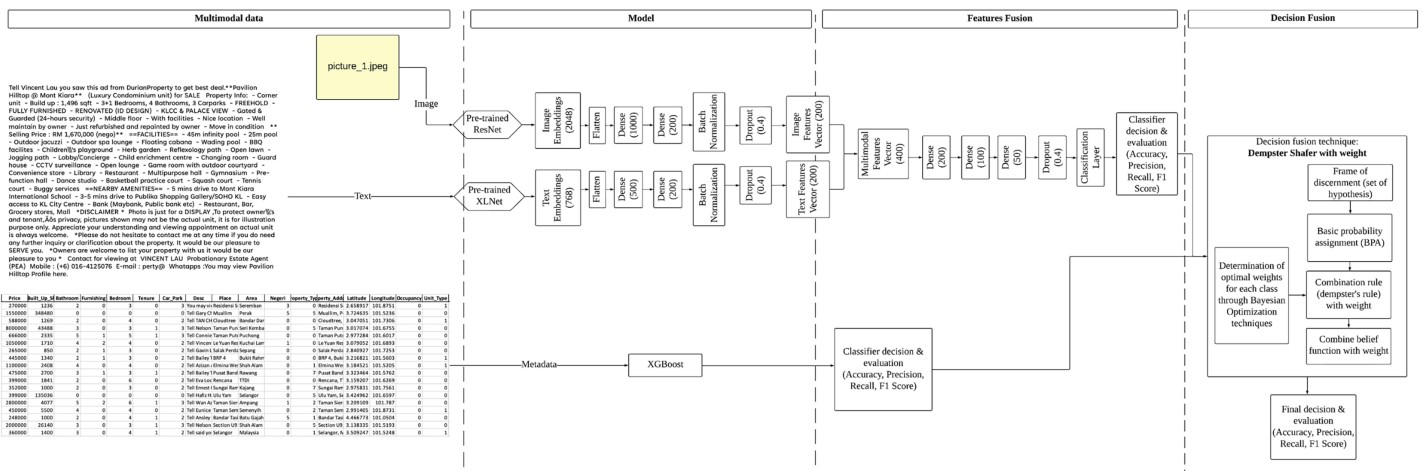

**Figure 1 A proposed CWDS-DLF model in the study.**

concurrently incorporating deep learning and machine learning in a single model when applying the early and late fusion method.

In the real estate domain, there are two studies that detect fraudulent real estate data based on property metadata. A study (*Nguyen-Duc, Nguyen & Nguyen, 2023*) utilized datasets from Vietnamese real estate websites and integrated multimodal machine learning with automated machine learning. By extracting textual data to obtain property metadata, this approach achieved a detection accuracy of 91.5% in identifying fraudulent advertisements, demonstrating the effectiveness of multimodal integration. Subsequently, another study (*Mohd Hamim & Sani, 2022*) focused on property metadata from two datasets, EdgeProp and DurianProperty, comprising over 124,000 records. Using K-Min clustering for preprocessing and support vector machine for classification, this study achieved a detection accuracy of 99.73% in identifying fraudulent real estate listings. These findings underscore the potential of advanced machine learning techniques, particularly multimodal approaches, in enhancing fraud detection systems in the real estate sector.

### Research gap

Overall, recent studies have introduced various innovative techniques and methods to improve the effectiveness of models in detecting false data. The development of these research models often uses multimodal data. New techniques, such as BLIP and CLIP, have been introduced to generate additional modalities.

Feature combination techniques, including cross-model attention, multimodal transformers, scaled dot-product attention, and modality-wise attention, have been utilized. At the decision level, techniques such as majority voting, weighted voting, Bayesian methods, neural networks, and Ensemble methods are employed.

All the techniques were found to obtain optimal performance for a balanced distribution of data for each class. However, achieving balanced multimodal data for 'fake' and 'not fake' classes in real estate data is difficult. Producing synthetic data is particularly complex when dealing with multimodal data, especially text-image-metadata pairs. Class

weighting is one alternative to help the model achieve optimal performance, but it must be integrated with other techniques to be effective.

An effective model that uses class weighting, called CWDS-DLF, is introduced to address this problem. This model employs combined techniques at both the feature and decision levels using deep learning and machine learning models. Its performance is further optimized by integrating Dempster–Shafer techniques and weights in the combined decision stage. Figure 1 illustrates the framework of the proposed CWDS-DLF model in the study. Please refer to this link for a clear diagram: https://github.com/maifuza/property-listings/tree/main.

# METHODS

This article proposes a novel multimodal method CWDS-DLF to enhance the fake property listings detection. Basically, this study will analyze image-text-metadata data pairs with two types of fusion, namely features and decision. The overall workflow of the model is shown in Fig. 2. The workflow consists of four main components: Data preparations, Labelling phase, Modelling- (Model selection, Features fusion, and Metadata classifier) and Decision fusion.

## Data preparations: data selection and preprocessing

To demonstrate the performance of this study, evaluations were conducted using property listings dataset, the online property listing in Malaysia. The dataset consists of 12,916 properties records in October 2021 such as metadata, texts and images of properties with 28 attributes. Figure 3 shows the sample of data used in this study. However, following the stages of cleaning, attribute generation and data transformation, only 17 attributes were retained for this study. The data is categorized into two classes: 'fake' and 'not fake'. Table 2 illustrates the distribution of data by class, while descriptions of the datasets can be found in Table 3. Four data preprocessing steps were carried out based on a study conducted previously (*Mohd Amin et al., 2024*):

(i) Cleaning: A statistical review of the data found some attributes with missing values such as 'NA' and −1. Attributes such as title (title associated with a property), facing (refers to the direction in which a property faces), land_D1, land_D2 (land_D1 and land_D2 was an address of property), postcode (postcode of property), agen_code (sales agent ID) and views (number of times a property listing has been viewed), which contained a value of −1, were classified as attributes without meaningful values and subsequently eliminated from the dataset. Additionally, certain attributes deemed unnecessary for this study, including url (the web link of the property listing), agent_name, date, and src (image source link), were also excluded from the analysis. Some records with missing values were replaced using information extracted from the Desc attribute, which consists of text data with unstructured content. To replace the missing values in the Car_Park attribute, the data for each housing type or Property_Type was analyzed to identify the most common Car_Park value. For instance, the value '2' was assigned to the Car_Park attribute for Terrace House

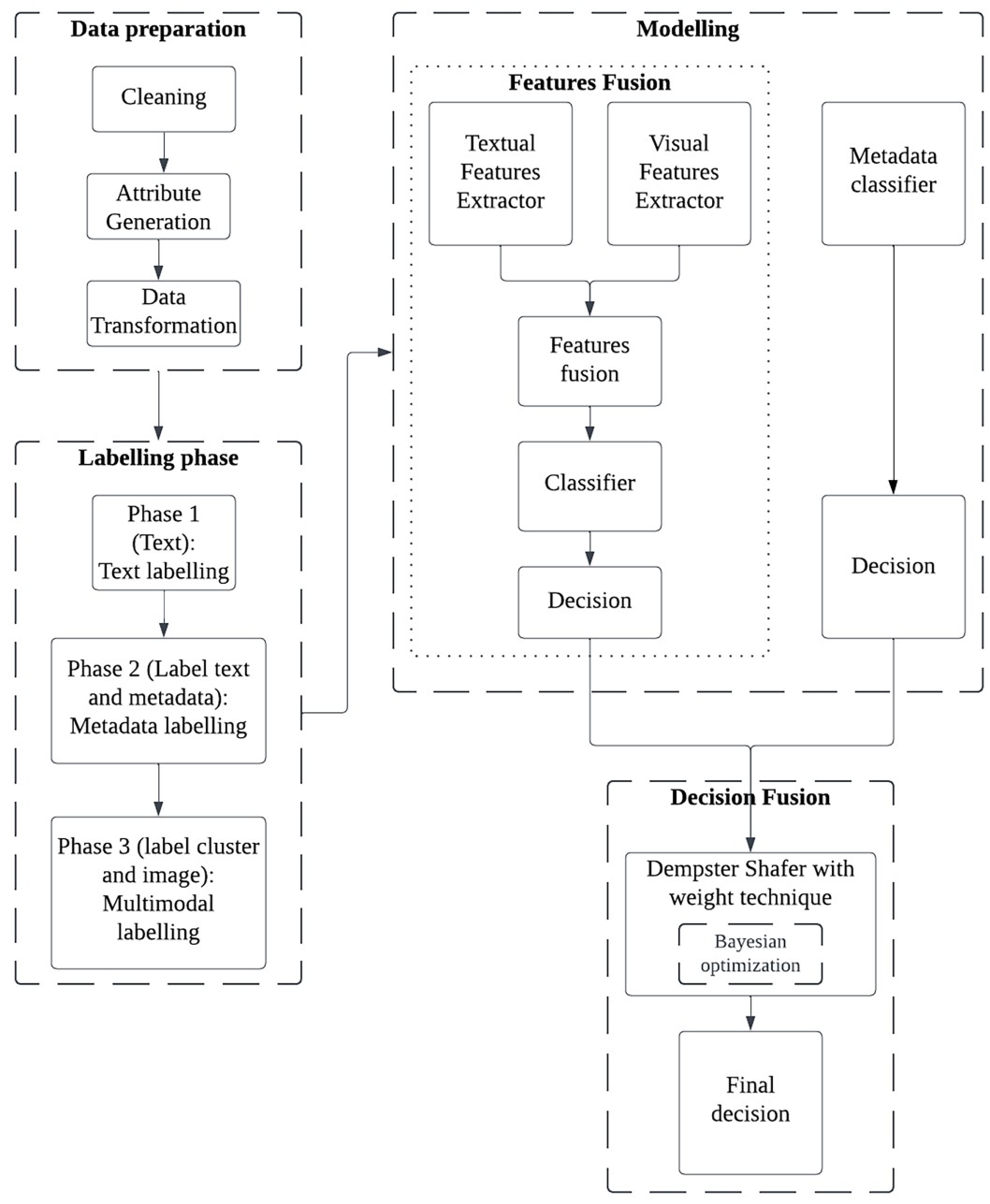

**Figure 2 The overall workflow for the CWDS-DLF model.**

properties, as this was the most frequently occurring value. A similar approach was used for the Bathroom and Bedroom attributes.

(ii) Attribute generation: Three attributes were generated: Longitude and Latitude, derived from the Address and Description attributes, and Expert_Label.

(iii) Data transformation: Most machine learning models require both input and output variables to be numerical (*Daud et al., 2023*). Therefore, all categorical or nominal attributes must be converted to numerical values before they can be used in grouping

Image data: This dataset comprises a variety of images depicting properties such as houses, land, shops, factories, and more.

Tell Vincent Lau you saw this ad from DurianProperty to get best deal.**Pavilion Hilltop @ Mont Kiara** (Luxury Condominium unit) for SALE Property Info: - Corner unit - Build up : 1,496 sqft - 3+1 Bedrooms, 4 Bathrooms, 3 Carparks - FREEHOLD - FULLY FURNISHED - RENOVATED (ID DESIGN) - KLCC & PALACE VIEW - Gated & Guarded (24-hours security) - Middle floor - With facilities - Nice location - Well maintain by owner - Just refurbished and repainted by owner - Move in condition ** Selling Price : RM 1,670,000 (nego)** ==FACILITIES== - 45m infinity pool - 25m pool - Outdoor jacuzzi - Outdoor spa lounge - Floating cabana - Wading pool - BBQ facilites - Children\\\'s playground - Herb garden - Reflexology path - Open lawn - Jogging path - Lobby/Concierge - Child enrichment centre - Changing room - Guard house - CCTV surveillance - Open lounge - Game room with outdoor courtyard - Convenience store - Library - Restaurant - Multipurpose hall - Gymnasium - Pre-function hall - Dance studio - Basketball practice court - Squash court - Tennis court - Buggy services ==NEARBY AMENITIES== - 5 mins drive to Mont Kiara International School - 3-5 mins drive to Publika Shopping Gallery/SOHO KL - Easy access to KL City Centre - Bank (Maybank, Public bank etc) - Restaurant, Bar, Grocery stores, Mall *DISCLAIMER * Photo is just for a DISPLAY ,To protect owner\\\'s and tenant‚Äôs privacy, pictures shown may not be the actual unit, it is for illustration purpose only. Appreciate your understanding and viewing appointment on actual unit is always welcome. *Please do not hesitate to contact me at any time if you do need any further inquiry or clarification about the property. It would be our pleasure to SERVE you. *Owners are welcome to list your property with us it would be our pleasure to you * Contact for viewing at VINCENT LAU Probationary Estate Agent (PEA) Mobile : (+6) 016-4125076 E-mail : perty@ Whatapps :You may view Pavilion Hilltop Profile here.

Text data: This dataset consists of textual descriptions detailing property information. The length of each text ranges from 300 to 500 words.

| Price | Built_Up_Sp | Bathroom | Furnishing | Bedroom | Tenure | Car_Park | Desc | Place | Area | Negeri | roperty_Typ | perty_Addr | Latitude | Longitude | Occupancy | Unit_Type |
|---|---|---|---|---|---|---|---|---|---|---|---|---|---|---|---|---|
| 270000 | 1236 | 2 | 0 | 3 | 0 | 3 | You may vie | Residensi S | Seremban | 3 | 0 | Residensi S | 2.658917 | 101.8751 | 0 | 1 |
| 1550000 | 348480 | 0 | 0 | 0 | 0 | 0 | Tell Gary Ch | Muallim | Perak | 5 | 5 | Muallim, P | 3.724635 | 101.5236 | 0 | 0 |
| 588000 | 1269 | 2 | 0 | 4 | 0 | 2 | Tell TAN CH | Cloudtree | Bandar Dar | 0 | 0 | Cloudtree, | 3.047051 | 101.7306 | 0 | 1 |
| 8000000 | 43488 | 3 | 0 | 3 | 1 | 3 | Tell Nelson | Taman Pun | Seri Kemba | 0 | 5 | Taman Pun | 3.017074 | 101.6755 | 0 | 0 |
| 666000 | 2335 | 5 | 1 | 5 | 1 | 3 | Tell Connie | Taman Puti | Puchong | 0 | 0 | Taman Puti | 2.977284 | 101.6017 | 0 | 0 |
| 1050000 | 1710 | 4 | 2 | 4 | 0 | 2 | Tell Vincent | Le Yuan Res | Kuchai Lam | 1 | 0 | Le Yuan Res | 3.079052 | 101.6893 | 0 | 0 |
| 265000 | 850 | 2 | 1 | 3 | 0 | 2 | Tell Gavin L | Salak Perda | Sepang | 0 | 0 | Salak Perda | 2.840927 | 101.7253 | 0 | 0 |
| 445000 | 1340 | 2 | 1 | 3 | 0 | 2 | Tell Bailey 1 | BRP 4 | Bukit Rahm | 0 | 0 | BRP 4, Buki | 3.216821 | 101.5603 | 0 | 1 |
| 1100000 | 2408 | 4 | 0 | 4 | 0 | 2 | Tell Azizan / | Elmina Wes | Shah Alam | 0 | 1 | Elmina Wes | 3.184521 | 101.5205 | 0 | 1 |
| 475000 | 2700 | 3 | 1 | 3 | 1 | 2 | Tell Bailey 1 | Pusat Band | Rawang | 0 | 7 | Pusat Band | 3.323464 | 101.5762 | 0 | 0 |
| 399000 | 1841 | 2 | 0 | 6 | 0 | 2 | Tell Eva Loc | Rencana | TTDI | 0 | 0 | Rencana, T | 3.159207 | 101.6269 | 0 | 0 |
| 352000 | 1000 | 2 | 0 | 3 | 0 | 2 | Tell Ernest ( | Sungai Ram | Kajang | 0 | 7 | Sungai Ram | 2.975831 | 101.7561 | 0 | 0 |
| 399000 | 135036 | 0 | 0 | 0 | 0 | 0 | Tell Hafiz H | Ulu Yam | Selangor | 0 | 5 | Ulu Yam, Se | 3.424962 | 101.6597 | 0 | 0 |
| 2800000 | 4077 | 5 | 2 | 6 | 1 | 3 | Tell Wan Az | Taman Sier | Ampang | 1 | 2 | Taman Sier | 3.209109 | 101.787 | 0 | 0 |
| 450000 | 5500 | 4 | 0 | 4 | 0 | 2 | Tell Eunice | Taman Sem | Semenyih | 0 | 2 | Taman Sem | 2.991405 | 101.8731 | 0 | 1 |
| 248000 | 1000 | 2 | 0 | 4 | 1 | 2 | Tell Ansley | Bandar Tasi | Batu Gajah | 5 | 1 | Bandar Tasi | 4.466773 | 101.0504 | 0 | 0 |
| 2000000 | 26140 | 3 | 0 | 3 | 1 | 3 | Tell Nelson | Section U9 | Shah Alam | 0 | 5 | Section U9, | 3.138335 | 101.5193 | 0 | 0 |
| 360000 | 1400 | 3 | 0 | 4 | 1 | 2 | Tell said yo | Selangor | Malaysia | 0 | 1 | Selangor, N | 3.509247 | 101.5248 | 0 | 1 |

Metadata: This dataset contains information on property details including price, property type, unit type, property area, location (state), number of bedrooms, bathrooms, and more.

**Figure 3** **The sample of data used in this study.**

**Table 2 The distribution of data by class.**

| Element | Classes | |
|---|---|---|
| | Fake (1) | Not fake (0) |
| Data distribution | 1,138 (17%) | 5,549 (83%) |
| Total | 6,687 | |

**Table 3 The descriptions of the datasets.**

| No | Attribute | Values | Label | Data Type |
|---|---|---|---|---|
| 1 | Price | Numbers | – | Float |
| 2 | Built_Up_Sf | Numbers | – | Float |
| 3 | Bathroom | Numbers | – | Integer |
| 4 | Property_Type | Condominium/Apartment/Flat/Serviced Residence | 0 | Nominal |
| | | Terrace House | 1 | |
| | | Link Bungalow/Semi-Detached House/Superlink | 2 | |
| | | Bungalow/Detached House/Villa | 3 | |
| | | Factory/Warehouse/Shop | 4 | |
| | | Agriculture/Bungalow Land/Commercial/Development Land | 5 | |
| | | Shop Office/SOHO/Office | 6 | |
| | | Townhouse/Cluster | 7 | |
| | | Hotel | 8 | |
| 5 | Occupancy | Vacant | 0 | Nominal |
| | | Tenanted | 1 | |
| | | Owner Occupied | 2 | |
| 6 | Unit_Type | Intermediate Lot | 0 | Nominal |
| | | Corner Lot | 1 | |
| | | End Lot | 2 | |
| 7 | Location | Selangor | 0 | Nominal |
| | | Kuala Lumpur | 1 | |
| | | Johor Baharu | 2 | |
| | | Negeri Sembilan | 3 | |
| | | Pulau Pinang | 4 | |
| | | Perak | 5 | |
| | | Kedah | 6 | |
| | | Cyberjaya | 7 | |
| | | Pahang | 8 | |
| | | Melaka | 9 | |
| | | Putrajaya | 10 | |
| | | Sabah | 11 | |
| | | Kelantan | 12 | |
| | | Terengganu | 13 | |
| | | Sarawak | 14 | |
| | | Perlis | 15 | |

(Continued)

| No | Attribute | Values | Label | Data Type |
|----|-----------|--------|-------|-----------|
| 8 | Furnishing | Unfurnished | 0 | Nominal |
| | | Partly Furnished | 1 | |
| | | Fully Furnished | 2 | |
| 9 | Bedroom | Numbers | – | Integer |
| 10 | Tenure | Leased hold | 0 | Nominal |
| | | Freehold | 1 | |
| 11 | Car_Park | Numbers | – | Integer |
| 12 | Cluster_Label | Not Fake | 0 | Nominal |
| | | Fake | 1 | |
| 13 | Longitude | Numbers | – | Float |
| 14 | Latitute | | | |
| 15 | Image | Image in .jpg format | – | – |
| 16 | Text | Description of the property | – | – |
| 17 | Text label | Not Fake | 0 | Nominal |
| | | Fake | 1 | |

and classification models. In this study, label (integer) encoding is applied to the property dataset, where each unique category value is converted to a distinct integer. This approach is suitable for tree-based models like XGBoost, as it efficiently handles categorical features without requiring One-Hot Encoding (*Gupta & Asha, 2020*). Since XGBoost uses decision trees to find optimal splits, it can process label-encoded values without assuming an ordinal relationship, ensuring both computational efficiency and model accuracy (*Hancock & Khoshgoftaar, 2020*). Table 3 lists these attributes along with their corresponding labels and data types.

(iv) Text preprocessing: Several preprocessing steps were applied to the real estate text data. These steps included the removal of URLs, reply to usernames, hashtags, punctuation, special characters, extra spaces, numbers, and stopwords. Additionally, the text was converted to lowercase, and character normalization was performed. The real estate text data used in this study is entirely in English. Subsequently, the preprocessed text was tokenized using the XLNet tokenizer.

## Labelling phase: multimodal real estate data labeling

The labeling of multimodal real estate data was conducted in three phases. The first phase involved textual data, where labels were obtained through the extraction of specific words from the text data identified by experts. The experts categorized the data as fake or not fake based on the textual content. These labels were then combined with real estate metadata during the second phase of labeling. In this phase, the K-Min clustering technique was employed to cluster expert labels and real estate metadata, generating cluster labels that classified the data as fake or not fake based on matching characteristics. The third phase involved matching fake and non-fake data with their corresponding image (only one image per data point). For example, data labeled as fake were verified to contain fake images. This

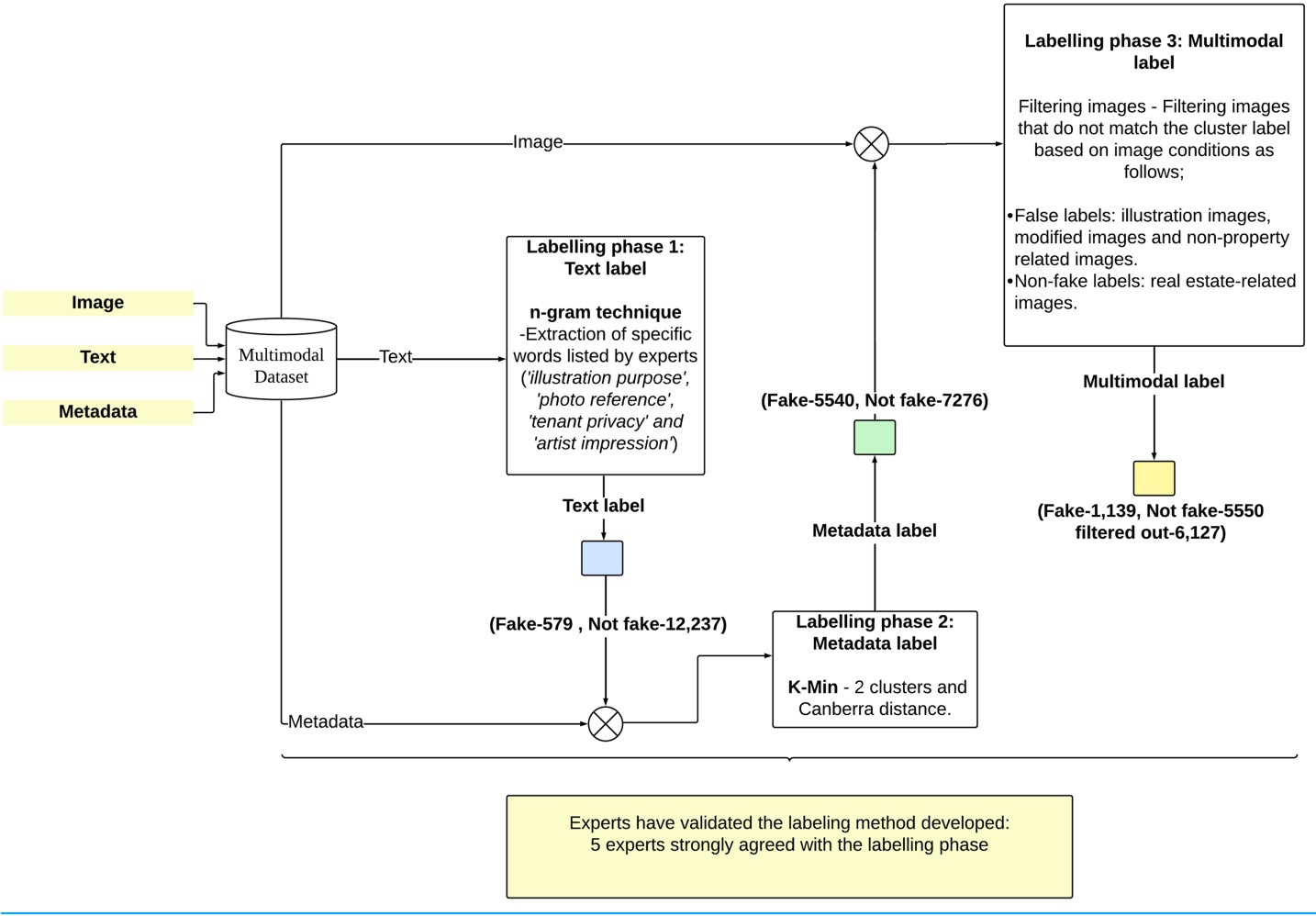

**Figure 4** A visual representation of the labelling phases.

process produced a comprehensive classification of fake and non-fake data, integrating textual data, metadata, and real estate images. Figure 4 provides a visual representation of the labelling phases. Finally, verification was conducted by five experts with extensive experience in real estate marketing to ensure the accuracy and reliability of each labeling phase.

### Labeling phase 1: text label

The first phase of labeling using textual data involved evaluating text based on specific words or terms such as 'illustration purposes', 'photo reference', 'tenant privacy', and 'artist impression', which were identified by experts as indicative of fraudulent characteristics (*Mohd Hamim & Sani, 2022*). These terms are often used to conceal deficiencies in real estate information by attaching incomplete or misleading property images. The n-grams technique was employed to detect these words in the text, where texts containing these terms were labeled as fake, while those without were labeled as not fake.

**Table 4 Multimodal label based on cluster and image labels.**

| Cluster label | Image label | Multimodal label |
|---|---|---|
| Fake | Fake | Fake (1,139 data) |
| Fake | Not Fake | The data will be filtered out (6,127) |
| Not Fake | Fake | |
| Not Fake | Not Fake | Not Fake (5,550 data) |

As a result, a total of 12,816 real estate data entries were classified into 579 not fake and 12,237 fake entries.

### Labeling phase 2: metadata label

The second labeling phase involved K-Min clustering analysis on real estate metadata consisting of 14 attributes (Price, Built_Up_SF, Bathroom, Property_Type, Occupancy, Unit_Type, Location, Furnishing, Bedroom, Tenure, Car_Park, Longitude and Latitude), including text labels from the first phase, encompassing 12,816 data records. Two clusters were formed, and the Canberra distance was selected as the centroid distance calculation method. Further details can be found in previous studies (*Mohd Amin et al., 2024*). The K-Min analysis resulted in the classification of 5,540 fake labels and 7,276 not fake labels.

### Labeling phase 3: multimodal label

This labeling phase represents the final stage, encompassing all modalities, namely text, metadata, and real estate images. The technique used in this phase is filtering, where images that do not correspond to their labels are removed from the dataset, as shown in Table 4. In this study, images are considered fake if they are generated using graphic design software, modified images, or images that do not represent actual real estate, such as pictures of trees, people, icons, and others (*CRES, 2024*; *Rodzi, 2015*; *Nasreen, 2024*). In contrast, not fake real estate images are those that represent actual properties.

Based on Table 4, a guideline was established, stating that real estate images must align with the cluster labels. For instance, modified images with a fake cluster label will be labeled as fake in the dataset. However, data entries will be removed if the images and cluster labels do not match. As a result, a total of 6,127 entries were filtered out to prevent errors in the detection model. As a result of the image-label matching process, 1,139 entries were labeled as fake, while 5,550 entries were labeled as not fake. These labels constitute the final dataset, which has been comprehensively filtered across multiple modalities, expert evaluations, and unique data patterns.

### Expert validation

Experts in the field of real estate are required to validate the accuracy and reliability of the labeling method employed. Consequently, five experts were selected to evaluate this labeling method. All the experts have over 10 years of experience in real estate marketing. Feedback from the experts was gathered through a questionnaire, as shown in Table 5.

**Table 5 Detailed survey items and expert responses to the developed labeling method.**

| Item | Quantity | Percentage |
| --- | --- | --- |
| Respondent demographics: | | |
| 1 Gender | 4 individuals - Male | 80% Male |
| | 1 individual - Female | 20% Female |
| 2 Years of experience in real estate transactions | 5 individuals - Over 10 years | 100% |
| 3 Area of expertise | 3 individuals - Real Estate Consultants | 60% |
| | 1 individual - Online Real Estate Platform Developer | 20% |
| | 1 individual - Geographic Information System Expert | 20% |
| Survey items: | | |
| 4 Agreement on the features of text data labeling | 5 experts strongly agreed | 100% Strongly Agree |
| 5 Agreement on the features of metadata labeling | 5 experts strongly agreed | 100% Strongly Agree |
| 6 Agreement on the features of image data labeling | 5 experts strongly agreed | 100% Strongly Agree |
| 7 Multimodal data is important for analysis in detecting fraudulent real estate listings | 5 experts strongly agreed | 100% Strongly Agree |
| 8 Multimodal data should be integrated into a single artificial intelligence model to develop an effective detection model for fraudulent real estate listings | 5 experts strongly agreed | 100% Strongly Agree |

Based on the questionnaire, all respondents strongly agreed with the three phases of labeling, indicating that the developed labeling method aligns with real-world conditions. Additionally, all respondents expressed strong agreement on the incorporation of multimodal data and integration methods in developing an effective model for detecting fraudulent real estate listings.

## Modelling: selection method

To select the optimal deep learning architectures for this study, we followed a systematic approach. This approach ensures that the chosen models and configurations were selected based on their task relevance, empirical performance, and computational efficiency. The following are the steps that have been taken:

### Initial model pool

We began by identifying a pool of widely used architectures for text and image data. For text, we considered models like BERT (*Devlin et al., 2018*) and XLNet (*Yang et al., 2020*) due to their strong performance in natural language understanding tasks. For image data, we evaluated convolutional neural network (CNN) models such as ResNet50, ResNet101, and ResNet152, which are known for their state-of-the-art results in image classification tasks (*He et al., 2016*).

### Task-specific criteria

The models were chosen based on their compatibility with the task at hand-fake real estate listings detection. For text, XLNet was selected over other models like BERT because of its ability to better capture long-range dependencies in textual data, which is critical for understanding nuanced property descriptions (*Yang et al., 2020*). For image processing,

**Table 6 The performance of ResNet50, ResNet101, and ResNet152 with XLNet.** The bold values indicate the best-performing results for each evaluation metric across the tested models.

| Model | Not Fake (0) | | | Fake (1) | | | Specificity | Recall | Precision | Accuracy | F1-score |
|---|---|---|---|---|---|---|---|---|---|---|---|
| | Precision | Recall | F1-score | Precision | Recall | F1-score | | | | | |
| ResNet50 & XLNet (200 features) | 0.93 | 0.91 | 0.92 | 0.62 | 0.69 | 0.65 | 0.62 | 0.93 | 0.91 | 0.88 | 0.92 |
| ResNet50 & XLNet (300 features) | 0.95 | 0.89 | 0.92 | 0.59 | 0.77 | **0.67** | 0.59 | 0.95 | 0.89 | 0.87 | 0.92 |
| ResNet50 & XLNet (400 features) | 0.93 | 0.93 | 0.93 | 0.68 | 0.67 | **0.67** | 0.68 | 0.93 | 0.93 | 0.89 | 0.93 |
| ResNet101 & XLNet (400 features) | 0.90 | **0.98** | **0.94** | **0.86** | 0.49 | 0.62 | **0.86** | 0.90 | 0.94 | **0.90** | **0.94** |
| ResNet152 & XLNet (400 features) | 0.95 | 0.87 | 0.91 | 0.55 | 0.80 | 0.65 | 0.55 | 0.95 | 0.87 | 0.86 | 0.91 |

ResNet101 was chosen over ResNet50 and ResNet152 because it balances depth and computational efficiency, making it ideal for detecting subtle visual clues in property images (*He et al., 2016*).

### Performance-based selection

After identifying candidate models, we evaluated them using a training set derived from our multimodal dataset. The dataset was split into 80% for training and 20% for testing, ensuring a balanced evaluation of model performance. Each model was then assessed using metrics such as precision, recall, F1-score, and accuracy. In this study, we systematically evaluated different model configurations to select the best-performing architecture for the fusion of image and text data. The models tested included ResNet50, ResNet101, and ResNet152 for image feature extraction, and XLNet for text. Table 6 summarizes the results of this evaluation, the bold values indicate the best-performing results for each evaluation metric across the tested models. ResNet101 and XLNet with 400 features consistently outperformed the other configurations, making it the optimal model choice for this study.

The model combination of ResNet101 and XLNet with 400 features emerges as the best performer among the evaluated architectures, cause to its balanced and consistent results in key performance metrics.

For the not fake class, the model demonstrates a strong performance with a precision of 0.90, recall of 0.98, and an F1-score of 0.94. This balance between precision and recall ensures that the model can reliably identify listings that are not fake, with minimal false positives. Comparatively, models such as ResNet50 and XLNet with 400 features perform similarly, but the ResNet101 and XLNet model has an advantage in terms of precision, which is critical for reducing false classifications.

The real strength of the ResNet101 and XLNet combination lies in its ability to detect the fake class, where it delivers a precision of 0.86, far surpassing other models in this category. While its recall for the fake class is quite lower at 0.49, the high precision is crucial in practical applications. In scenarios where the goal is to minimize false alarms, high precision is preferred to ensure that detected fake listings are truly fraudulent. In contrast, other models may offer higher recall but often at the cost of lower precision, leading to a higher number of false positives, which is less desirable.

Additionally, this model maintains a specificity of 0.86, which further strengthens its ability to correctly identify true negatives (not fake listings), ensuring that the model doesn't over-classify listings as fake. This specificity figure, combined with an overall accuracy of 0.90 and the highest F1-score of 0.94 among the tested models strengthen the ResNet101 and XLNet combination as the most balanced and effective model for this task.

In conclusion, ResNet101 and XLNet with 400 features is superior due to its high precision in detecting fake listings, strong accuracy, and well-balanced performance across precision, recall, and specificity. This combination provides the best trade-off between catching fake listings and minimizing false positives, making it the optimal choice for the given multimodal task.

## Modelling: feature fusion strategy

Text and image modalities are separately embedded using models. Subsequently, the embedded vectors from text and image are fused together before being fed into several neural network layers to make a decision.

### Textual feature extractor-XLNet

The text descriptions in property listings consist of several sentences. Many text-based fake detection methods rely on traditional word vector models, which excel in analyzing straightforward sentences. However, for precise extraction of textual features, we utilize the pre-trained XLNet model as the core feature extractor. This choice is motivated by XLNet's ability to capture richer contextual information through its permutation language model architecture (*Athira et al., 2022*).

The textual feature input comprises a series of sentences arranged sequentially, obtained by segmenting the entire text using a sentence tokenizer (*Aziz, Bakar & Yaakub, 2024*). This segmentation produces a set of processed sentences, denoted as $N_t = \{S_1, S_2, \ldots S_n\}$, where $N_t$ represents the compiled sentences, $n$ indicates the count of sentences, and each $S_i$ represents a segment of the news article. These processed sentences are then fed into a pre-trained XLNet model for feature extraction, producing textual embeddings corresponding to the input. The resultant embeddings encapsulate the overall textual content of the article, as defined in Eq. (1).

$$E_t = \{E_1, E_2 \ldots E_i\} = XLNet(N_t). \tag{1}$$

The set $E_t$ encompasses textual embeddings extracted from the news content. Each sentence contributes a hidden representation denoted as $E_i$, produced by a pre-trained *XLNet* model. Given the diverse lengths of news articles, the length of the textual embedding is standardized. To ensure high-quality embeddings, the textual feature extractor utilizes an XLNet model pre-trained on a variety of language datasets. For this research, the Durian Property dataset is employed, focusing on English language data. Consequently, the pre-trained XLNet-base-cased model trained on English corpora is loaded.

### Visual feature extractor-ResNet101

Recognizing that images carry crucial information for distinguishing fake property listings, a visual feature extractor will be established to handle feature extraction. This article utilizes the ResNet101 model as the primary component of the visual feature extractor with the objective of extracting features from images. This convolutional network architecture has been pre-trained on the ImageNet database (*Ying et al., 2021*). Initially, the input image undergoes pre-processing, such as resizing and normalization, to ensure compatibility with the network. Then, the image is fed through the ResNet101 model, consisting of multiple layers including convolutional layers and residual blocks, which extract hierarchical features capturing various visual patterns. As the image progresses through the network, these features become increasingly abstract and representative of the image's content. Following the convolutional layers, global average pooling (GAP) is applied to aggregate spatial information into a single vector, constituting the embedding. This embedding effectively encapsulates the image's visual characteristics in a compact numerical format. The formula for *GAP* is defined by Eq. (2) typically involves taking the average of each feature map along its spatial dimensions.

$$GAP(X)_i = \frac{1}{H \times W} \sum_{j=1}^{H} \sum_{k=1}^{W} X_{ijk} \tag{2}$$

where $X$ is the input feature map, $H$ and $W$ are the height and width dimensions, respectively, and $X_{ijk}$ represents the value at position $(i,j,k)$ in the feature map. Optionally, normalization techniques may be applied to the embedding vector. The resulting image embedding can serve as an input alongside text input for subsequent classification tasks.

### Features fusion layer

Furthermore, the embedded vectors obtained from two distinct models, XLNet for text and ResNet101 for images, undergo processing through several neural network layers, including flatten, dense, batch normalization, and dropout. It is well known that neural network layers are responsible for processing and transforming input data to generate meaningful outputs (*Raj & Meel, 2021*).

The flatten layer reshapes multi-dimensional input data into a one-dimensional array, facilitating compatibility with subsequent layers such as dense layers. Dense layers, also known as fully connected layers, connect each neuron in one layer to every neuron in the next layer, enabling complex feature learning and representation (*Chen et al., 2022*). Batch normalization normalizes the activations of each layer within mini-batches during training, improving convergence speed and overall stability by reducing internal covariate shift. Dropout, a regularization technique, randomly deactivates neurons in a layer during training to prevent overfitting by encouraging the network to learn more robust features. These layers collectively contribute to the learning and generalization capabilities of neural networks across various tasks.

Hence, after passing through multiple layers of neural networks, the embedded feature vectors of the image and text are merged into a concatenated embedded features vector.

Subsequently, this concatenated vector progresses through the dense layer until a classification decision is reached, determining whether the input is categorized as fake or not fake.

## Modelling: metadata classifier-XGBoost

XGBoost is utilized to analyze the metadata for property listings due to several advantages. Firstly, XGBoost is renowned for its exceptional performance in handling structured/tabular data, making it particularly well-suited for tasks involving metadata analysis. Since property listings typically consist of structured data such as price, location, area, and features, XGBoost's ability to handle such data efficiently ensures accurate and reliable analysis. Additionally, XGBoost is highly scalable and can handle large datasets with ease, which is crucial when dealing with extensive property listings databases. Moreover, XGBoost's robustness against overfitting and its capability to handle missing data effectively further solidify its suitability for this task. Overall, XGBoost's combination of performance, scalability, and robustness makes it an excellent choice for analyzing property listing metadata, providing valuable insights for decision-making in the real estate domain (*Zhao, Chetty & Tran, 2019*).

## Decision fusion strategy

This stage is pivotal as it focuses on refining the fusion technique to optimize the performance of the model. Since the fusion of results serves as the final fusion in this model, there is a need for improvements on the basic technique (Dempster–Shafer) so that the overall effectiveness of the model can be improved.

### Decision fusion in Dempster–Shafer

The Dempster–Shafer theory, also known as evidence theory or belief function theory, is a mathematical framework used for reasoning under uncertainty and combining evidence from multiple sources. In decision fusion, this theory offers a principled approach to integrate information from various sources to make informed decisions.

Dempster–Shafer theory operates on the concept of belief functions, which quantifies the uncertainty associated with different pieces of evidence. Each piece of evidence is represented by a belief function, which assigns a degree of belief to each possible hypothesis or outcome (*Freedman, 1994*; *Zhang, Sjarif & Ibrahim, 2022*).

The key aspect of Dempster–Shafer theory is the combination rule, also known as Dempster's rule of combination, which computes the belief in each hypothesis by combining the individual belief functions. This combination takes into account not only the evidence supporting a hypothesis directly but also the evidence that indirectly supports it through its relationship with other hypotheses.

The Dempster–Shafer theory is particularly useful in decision fusion scenarios where there are multiple sources of evidence, each with its own uncertainties and reliability levels. By combining evidence using Dempster's rule, the resulting belief functions provide a comprehensive and coherent representation of uncertainty, allowing for more robust decision-making in complex and uncertain environments (*Somero, Snidaro & Rogova, 2022*).

### Class weighted Dempster–Shafer

Enhancing Dempster–Shafer techniques with class weights can lead to significant improvements in decision-making processes. Class weights allow the algorithm to assign different levels of importance to different classes or hypotheses, effectively addressing imbalances in the dataset. By assigning higher weights to minority classes or hypotheses, the algorithm becomes more sensitive to their importance, thereby improving its ability to make accurate predictions or decisions for these classes.

The incorporation of class weights into Dempster–Shafer techniques helps to mitigate the effects of class imbalance, which is a common challenge in many real-world applications. This enhancement ensures that the algorithm pays more attention to less frequent classes or hypotheses, preventing them from being overshadowed by dominant ones. As a result, decision-making becomes more balanced and robust, leading to better overall performance and effectiveness of the Dempster–Shafer fusion technique.

In Dempster–Shafer theory, combining belief and measuring conflict are essential steps for reasoning under uncertainty and making decisions based on evidence from multiple sources. Combined belief in Eqs. (3) and (4) refers to the integration of evidence from multiple sources to determine the overall belief in a hypothesis or proposition. In Dempster–Shafer theory, this combination is achieved using Dempster's rule of combination. When evidence is available from multiple sources, each providing belief functions (also known as mass functions), Dempster's rule is applied to combine these belief functions into a single belief function representing the overall belief in each hypothesis or proposition. For a given hypothesis $H$, if we have belief functions $m_1$ and $m_2$ from two sources, the combined belief function $m$ is calculated in Eqs. (3) and (4).

$$m(A) = \frac{\sum_A w_1 m_1(A) m_2(A)}{1 - K} \tag{3}$$

$$m(B) = \frac{\sum_B w_2 m_1(B) m_2(B)}{1 - K} \tag{4}$$

$m(H)$ is the combined belief in hypothesis $H$. $m_1(A)$ and $m_2(B)$ are the belief values assigned by the first and second sources to subsets $A$ (class fake) and $B$ (class not fake) of the frame of discernment, respectively. $w_1$ and $w_2$ are the weight for the two classes. $K$ is the conflict measure defined by Eq. (5).

$$K = [w_1 \times m_1(A) \times m_2(B)] + [w_2 \times m_1(B) \times m_2(A)]. \tag{5}$$

Conflict measure ($K$) quantifies the degree of inconsistency or conflict between the pieces of evidence provided by different sources. In Dempster–Shafer theory, conflict arises when evidence from different sources does not completely agree, leading to uncertainty in decision-making. Conflict measure is a crucial factor in Dempster's rule because it affects the normalization of the combined belief function. A high conflict measure indicates significant discrepancy or inconsistency between the evidence provided by different sources, which may lead to higher uncertainty in decision-making (*Somero, Snidaro & Rogova, 2022*).

Overall, the integration of class weights into Dempster–Shafer techniques provides a mechanism to address class imbalances, leading to more accurate and reliable decision-making processes. This improvement enhances the algorithm's ability to handle diverse datasets and improves its overall effectiveness in various applications.

(i) Class weight determination method

The appropriate weighting value is determined using the Bayesian Optimization technique. It is a powerful technique for hyperparameter tuning, which is the process of finding the best configuration of parameters for a machine learning model to optimize its performance on a given task. It offers several advantages over traditional grid search or random search methods. It intelligently selects hyperparameter configurations to evaluate, leading to faster convergence to optimal or near-optimal configurations with fewer evaluations. Additionally, it can handle noisy objective functions and is more efficient in high-dimensional search spaces. Overall, Bayesian optimization is a popular and effective approach for hyperparameter tuning in machine learning.

## EXPERIMENT SETUP

The experiment was conducted using a computing infrastructure equipped with an Apple M2 chip, running on macOS Ventura version 13.4. The system was configured with 8 GB of memory. The code and datasets used in this study are publicly available at https://github.com/maifuza/property-listings/tree/main in GitHub.

### Features fusion

The feature extraction layer processes textual content and associated images to complete the extraction steps. In the extraction of textual feature representations, a pre-trained XLNet-base-cased model is employed to ensure the quality of the extracted features, given that the datasets used in the experiments are in English. The embedded vector features extracted for text input consist of 768 dimensions. Following this, the embedded vector features are passed through a dense layer with 500 dimensions, which subsequently reduces them to 200 dimensions. The ReLU activation function and L2 regularization with a value of 0.01 are applied to both layers. Afterward, batch normalization and dropout layers with a parameter of 0.4 are applied.

To extract visual feature representations, this article utilizes the ResNet101 convolutional network, pre-trained on ImageNet. The embedded vector features extracted for the input image consist of 2,048 dimensions. Subsequently, these embedded vector features are reduced to 200 dimensions through a dense layer with 1,000 dimensions, followed by another dense layer with 200 dimensions. The ReLU activation function and L2 regularization with a value of 0.01 are applied to both layers. Finally, batch normalization and dropout layers with parameter 0.4 are applied.

To integrate image and text feature representations, the features are combined into a 400-dimensional vector. Subsequently, this concatenated feature undergoes processing

**Table 7 The tuning value set for the analysis model.**

| Modality | Model extractor | Neural network layer setup (best setup) |
|---|---|---|
| Image | ResNet101 | Embedding vector (2,048 dimensions) |
| | | Flatten |
| | | Dense vector (1,000 dimensions) |
| | | Dense vector (200 dimensions) |
| | | Batch normalization |
| | | Dropout (0.4) |
| | | Image features (200) |
| Text | XLNet | Embedding vector (768) |
| | | Flatten |
| | | Dense (500) |
| | | Dense (200) |
| | | Batch normalization |
| | | Dropout (0.4) |
| | | Text features (200) |
| Multimodal (Image and text) | ANN | Multimodal features (400) |
| | | Dense (200) |
| | | Dense (100) |
| | | Dense (50) |
| | | Dropout (0.4) |
| | | Classification layer (2) |

through a neural network structure comprising a fully connected layer employing the ReLU activation function, followed by a classification layer utilizing the SoftMax activation function. The fully connected layer parameters include L2 regularization with a coefficient value of 0.01, and its dimensions are set to 400, 150, and 50. To prevent overfitting during training, the model implements an early stopping mechanism. Table 7 outlines the specific tuning values employed for the analysis model. The best value of the hyperparameters is obtained after tuning with several other potential values.

## Metadata classifier

Hyperparameter tuning for the XGBoost model applied to metadata analysis was performed manually. Initial parameters, such as learning rate (Lr), number of trees (#tree), tree depth (tree_depth), subsample ratio (sub_samp), and column sample ratio (col_samp), were selected based on prior research and practical guidelines for XGBoost (*Bentéjac, Csörgő & Martínez-Muñoz, 2021*; *Putatunda & Rama, 2019*). The hyperparameters were then manually adjusted, as shown in Table 8. The learning rate was tested with values ranging from 0.01 to 0.3, while the tree depth was adjusted from 2 to 10 to balance model complexity and overfitting. Similarly, the subsample and column sampling ratios were varied from 0.6 to 1.0 to prevent overfitting while maintaining adequate feature sampling. For each iteration, the model was trained and evaluated on the

**Table 8 Tuned hyperparameter values in XGBoost.**

| Parameter | Range of tunning value | Best value |
|---|---|---|
| Learning Rate (Lr) | 0.01–0.3 | 0.1 |
| Number of Trees (#tree) | 100–500 | 300 |
| Tree Depth (tree_depth) | 2–10 | 8 |
| Subsample (sub_samp) | 0.5–1 | 0.8 |
| Column Subsampling (col_samp) | 0.5–1 | 0.8 |
| Learning Task Parameters | Multi Softmax | |

**Table 9 Performance of the XGBoost model based on various tuned hyperparameter values.** The bolded values indicate the best performance scores corresponding to specific hyperparameter settings.

| Tuned value | Not Fake (0) | | | Fake (1) | | | Specificity | Recall | Precision | Accuracy | F1-score |
|---|---|---|---|---|---|---|---|---|---|---|---|
| | Precision | Recall | F1-score | Precision | Recall | F1-score | | | | | |
| Lr=0.01, #tree=100, tree_depth=2, sub_samp=0.6, col_samp=0.6 | **0.98** | **0.98** | 0.92 | **0.80** | 0.42 | 0.55 | **0.80** | 0.89 | **0.98** | 0.88 | **0.93** |
| Lr=0.05, #tree=200, tree_depth=4, sub_samp=0.7, col_samp=0.7 | 0.93 | 0.93 | **0.93** | 0.66 | **0.65** | **0.66** | 0.66 | **0.93** | 0.93 | 0.88 | **0.93** |
| Lr=0.1, #tree=300, tree_depth=6, sub_samp=0.8, col_samp=0.8 | 0.93 | 0.93 | **0.93** | 0.65 | 0.64 | 0.65 | 0.65 | **0.93** | 0.93 | 0.88 | **0.93** |
| Lr=0.2, #tree=400, tree_depth=8, sub_samp=0.9, col_samp=0.9 | 0.92 | 0.93 | 0.92 | 0.64 | 0.60 | 0.62 | 0.64 | 0.92 | 0.93 | 0.87 | 0.92 |
| Lr=0.3, #tree=500, tree_depth=10, sub_samp=1.0, col_samp=1.0 | 0.91 | 0.94 | **0.93** | 0.66 | 0.57 | 0.61 | 0.66 | 0.91 | 0.94 | 0.87 | **0.93** |
| Lr=0.1, #tree=300, tree_depth=8, sub_samp=0.8, col_samp=0.8 | 0.92 | 0.94 | **0.93** | 0.69 | **0.65** | **0.66** | 0.69 | **0.93** | 0.94 | **0.89** | **0.93** |

validation dataset using performance metrics such as accuracy, precision, recall, and F1-score.

Based on Table 9, the optimal tuning values for the XGBoost model specifically a learning rate of 0.1, 300 trees, tree depth of 8, sub-sampling rate of 0.8, and column sampling rate of 0.8 lead to the best performance across various metrics. These values result in the highest overall accuracy and F1-score across the tested configurations, making them ideal for detecting both fake and not fake classes in this task. The bolded values in Table 9 indicate the best performance scores corresponding to specific hyperparameter settings.

The learning rate of 0.1 strikes an effective balance between model convergence speed and performance. A smaller learning rate, such as 0.01, would slow down the learning process and risk underfitting the model. Conversely, a larger learning rate (*e.g.*, 0.2) could result in overfitting due to excessively fast convergence. Thus, the selected rate of 0.1 ensures that the model converges at a steady pace, achieving optimal performance without sacrificing generalization.

In terms of the number of trees, 300 provides sufficient model complexity while avoiding overfitting. Increasing the number of trees beyond 300 does not significantly

improve performance; in fact, higher tree counts, such as 400 or 500, can result in diminishing returns and increased computational costs. The tree depth of 8 is another key factor contributing to the model's success, allowing it to capture complex patterns in the data. Deeper trees (*e.g.*, depth of 10) lead to overfitting, while shallower trees may not capture enough complexity, especially in tasks like fake detection where subtle distinctions are important.

The sub-sampling rate of 0.8 further enhances the model's robustness by ensuring that only 80% of the training data is used for each tree, preventing overfitting. This introduces randomness into the model without losing too much information, as would happen with lower sub-sampling rates (*e.g.*, 0.6). Similarly, the column sampling rate of 0.8 ensures that each tree is built using a subset of 80% of the features, adding further diversity and helping the model generalize better by reducing correlation between trees.

Finally, the combination of these tuning parameters produces a model that balances precision, recall, and overall accuracy effectively. This XGBoost configuration delivers the highest accuracy (0.89) and F1-score (0.93), making it the most suitable setup for this task.

## Decision fusion

In decision fusion, two input sources are involved. The first input is obtained from the combined feature classification results. The second input comes from a metadata classifier using the XGBoost analysis model. The prediction results from both classifiers are combined using the weighted Dempster–Shafer technique. The weighted values for the 'fake' and 'not fake' classes are integrated into the basic Dempster–Shafer method to produce more optimal classification performance, taking into account the data distribution imbalance between classes. Figures 5 and 6 provides a detailed view and pseudo code of the weighted Dempster–Shafer integration technique. The technique is conducted as follows: Optimal weight determination: The optimal weights $\widetilde{w_{nf}}$ and $\widetilde{w_f}$ are typically found using a technique such as Bayesian optimization. This process involves tuning the weights to maximize classification performance, taking into account the imbalance between classes. Conflict measure calculation: The conflict measure $K$ quantifies the degree of disagreement between the classifiers for different classes. This measure is crucial as it adjusts the influence of combined beliefs in the final decision. Combined belief calculation: Beliefs for each class are calculated by considering the probabilities from both classifiers, weighted by the optimal weights. The denominator $1 - K$ ensures normalization by accounting for conflicts. Class justification: The class with the higher combined belief is selected as the final classification. This step ensures that the decision reflects the most probable class based on the combined evidence.

The Bayesian optimization technique is used to determine the weighted values. This technique helps in assigning more accurate values for each class. As a result, the weighting value for the 'fake' class is 1.0, and for the 'not fake' class, it is 0.1. Figure 7 illustrates how these weighting values are determined in detail. The technique is conducted as follows: Bayesian optimization: A loss function $\alpha(\omega)$ is defined to evaluate the performance of each weight. The weight that maximizes this loss function is identified as the optimal weight using Bayesian optimization techniques. Bayesian optimization uses a

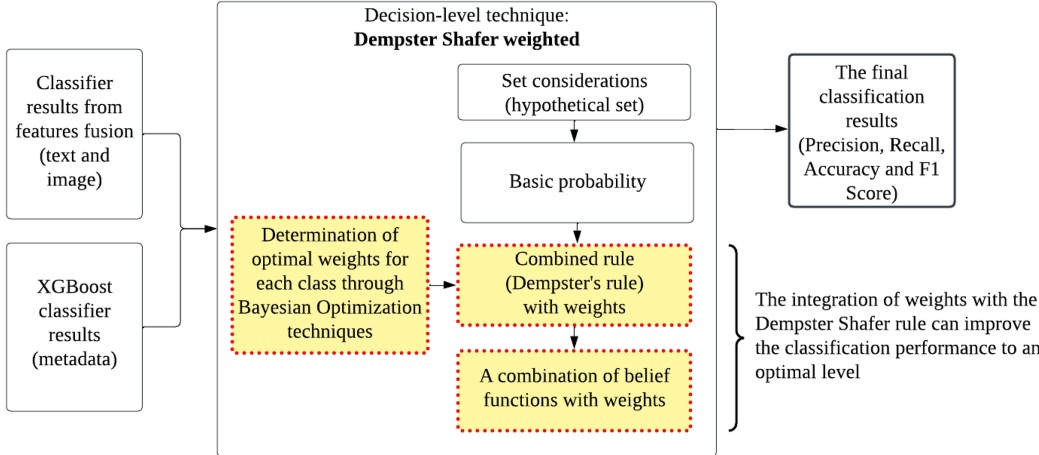

**Figure 5** A view of the weighted Dempster–Shafer integration technique.

## Decision fusion using Dempster Shafer with class weight technique

1. **Input**: Optimal weight $\widetilde{w}_{nf}, \widetilde{w}_f$ where $nf$ is not fake, $f$ is fake
   Probability $P_{a,nf}, P_{a,f}, P_{b,nf}, P_{b,f}$ where $a$ is first classifier and $b$ is second classifier
2. **Output**: Class fake or not fake
3. **Process**:
4. Find the optimal weight $\widetilde{w}_i$ where $i \in \{f, nf\}$ (refer pseudo code for optimal weight)
5. Determine conflict measure $K$:
6. $K \leftarrow \left(\widetilde{w}_{nf} \times P_{a,f} \times P_{b,nf}\right) + (\widetilde{w}_f \times P_{a,nf} \times P_{b,f})$
7. Calculate combine belief M:
8. $M_i \leftarrow \frac{\widetilde{w}_i \times P_{a,i} \times P_{b,i}}{1-K}$ where $i \in \{f, nf\}$
9. Justify class:
10. **if** $M_f > M_{nf}$
11.     $class \leftarrow fake$
12. **else**
13.     $class \leftarrow not\ fake$
14. End

**Figure 6** Pseudocode for decision fusion using Dempster–Shafer with the class weight technique.

probabilistic model to suggest the most promising weights to test next, balancing exploration and exploitation. Training and evaluation: The optimal weight is used to train a logistic regression model. The probability of class membership is computed using the logistic function. The model's accuracy is evaluated based on its classification performance. Final output: The optimal weight that yields the highest classification accuracy is produced.

**Optimal weights for each class using Bayesian Optimization technique**

1. **Input**: Weight $w_i$ for class $i$
2. **Output**: Optimal weight $\widetilde{w}_i$ for class $i$
3. **Process**:
4. Define a range of candidate weights:
5.     $w_i \leftarrow [w_{i,1}, w_{i,2}, w_{i,3} \ldots \ldots w_{i,n}]$ where $i \in \{f, nf\}$, $nf$ is not fake, $f$ is fake
6.     Each candidate weight:
7.     $w_{i,j} \leftarrow \frac{j}{n}$ where $j \in (1,2,3, \ldots \ldots, n)$
8. Optimize the weights using Bayesian optimization:
9.     Define a loss function: $\alpha(w)$
10.     Find the optimal weight $\widetilde{w}_i$ of $w_i$ by maximizing $\alpha(w)$:
11.     $\widetilde{w}_i \leftarrow \arg\max_{w_i} \alpha(w)$
12. Train a Logistic Regression model using the optimal weights $\widetilde{w}_i$:
13.     Compute the probability $P(y \leftarrow 1|X)$ with:
14.     $P(y \leftarrow 1|X) \leftarrow \frac{1}{1+e^{-(X \cdot \widetilde{w}_i + b)}}$
15.     where X is input features, y is label, $\widetilde{w}_i$ is optimal weight and b is the bias term
16. Calculate the accuracy of the Logistic Regression model:
17. Measure the accuracy as: $accuracy \leftarrow \frac{Number\ of\ correctly\ classified\ examples}{Total\ number\ of\ examples}$
18. Output the optimal weight: $\widetilde{w}_i$ for class $i$
19. End

**Figure 7 Pseudocode for defining class weight using Bayesian optimization.**

## Evaluation metrics

In fake news detection tasks, accuracy and F1-score are standard metrics for evaluating model (*Dixit, Bhagat & Dangi, 2022*; *Mathews & Preethi, 2022*; *Padalko, Chomko & Chumachenko, 2024*). In this experiment, we thoroughly evaluate the model's performance by also considering specificity, precision, and recall as additional metrics. These metrics are defined by Eqs. (6) to (9). For all these indicators, higher values indicate better model performance.

$$Accuracy = \frac{TP + TN}{TP + TN + FP + FN} \tag{6}$$

$$Precision = \frac{TP}{TP + FP} \tag{7}$$

$$Recall = \frac{TP}{TP + FN} \tag{8}$$

$$F1 - score = \frac{2 \times TP}{2 \times TP + FN + FP}. \tag{9}$$

*TP* denotes the number of actual positive cases correctly predicted as positive. *FP* refers to the number of actual negative cases incorrectly predicted as positive. *FN* indicates the number of actual positive cases mistakenly predicted as negative. *TN* represents the number of actual negative cases correctly predicted as negative. In addition to accuracy, we also evaluated model performance using the area under the curve (AUC) metric, which is

commonly used to measure classification effectiveness, particularly in imbalanced datasets. AUC measures the ability of a model to distinguish between classes by summarizing the receiver operating characteristic (ROC) curve into a single value. Higher AUC indicates a better model at distinguishing between positive and negative classes. For example, an AUC value between 0.8 and 0.9 indicates good discrimination, while an AUC value between 0.9 and 1.0 indicates excellent discrimination.

## Baseline methods

The baseline method is applied to real estate data based on the suitability of the data structure, which includes images, text, and metadata.

### Single-modal approaches

CNN (*Chen et al., 2022*) are widely used in various fields, especially in computer vision tasks like image recognition, object detection, and image classification. They are designed to automatically and adaptively learn spatial hierarchies of features from input images.

XLNet (*Athira et al., 2022*; *Liang, 2023*) has demonstrated state-of-the-art performance on a wide range of natural language processing tasks, surpassing previous models like BERT in many cases. Its ability to capture bidirectional context effectively during training makes it particularly well-suited for understanding and generating natural language text.

XGBoost (*Zhao, Chetty & Tran, 2019*) is a versatile and effective algorithm that has become a popular choice for machine learning competitions and real-world applications due to its high performance and robustness.

ResNet (*Ilhan, Serbes & Aydin, 2022*; *Ying et al., 2021*; *Zhou et al., 2022*) is a CNN architecture widely utilized as a feature extractor in various tasks, particularly in the field of image classification.

Random forest (*Mohd Amin et al., 2024*) was employed to evaluate the clustering outcomes in this study, where a K-means clustering model was developed to distinguish between genuine and fraudulent property listings.

### Multi-modal approaches

In the multimodal approach, several baseline methods are employed on real estate data to evaluate the performance of both the existing study model and the proposed one.

In the adaptation of the Multimodal Fake News Detection (MMFND), model extractor ResNet (image) and XLNet (text) are employed to encode both the text and image, which are then concatenated to create the final embedding. This embedding is then fed into a multimodal transformer for classification (*Liu et al., 2023*).

Spotfake+ also employed the same model extractor in MMFND for image and text, but applying the artificial neural network to detect the fake news (*Singhal et al., 2020*).

A study utilized the Evidential Transferable Belief Model (TBM) to combine classifiers for COVID diagnosis (*Somero, Snidaro & Rogova, 2022*). Using the property data, ANN is employed to classify image and text data, while XGBoost is utilized for property metadata classification. Following the computation of classification decisions by each model, the Dempster–Shafer fusion is applied. Finally, another Pignistic probability is employed for the final classification.

The detection of fraudulent real estate advertisements using Automated Multimodal Learning (FADAML) incorporates property data such as price, area, road, and district extracted from textual data. FADAML integrates multiple machines learning models, including categorical boosting, extreme gradient boosting, random forest, extra tree classifier, and K-nearest neighbor, to achieve more accurate classification.

Furthermore, we also incorporated other combined techniques such as Majority Voting (*Ilhan, Serbes & Aydin, 2022*), Weighted Voting (*Gumaei et al., 2022*), and Dempster–Shafer (*Oh & Kang, 2017*) to compare them with the proposed model.

## RESULTS DISCUSSION

The results presented in Table 10 offer a thorough analysis of the proposed model's performance across various evaluation metrics, providing valuable insights into its effectiveness in detecting fake property listings. The bolded values in the table represent the best-performing results across the tested configurations.

Unimodal analysis, focusing on individual modalities such as text, image, or metadata, reveals subtle performance variations among different models. For example, the ResNet101 model demonstrates high precision of 0.91 and recall of 1.00 for identifying genuine content but struggles with fake content detection, resulting in a low F1-score of 0.02 in fake content. Similarly, the 2D CNN model exhibits strong precision of 0.90 and recall of 0.99 for genuine content but faces challenges in identifying fake content, leading to imbalanced precision, recall metrics and a low F1-score of 0.02. A similar trend is observed with the Random Forest model, which achieves an F1-score of only 0.42 for fake content. These findings highlight the limitations of unimodal approaches for accurately discerning fraudulent property listings.

In contrast, the multimodal analysis by integrating information from multiple modalities, shows improved performance across various metrics as compared to unimodal methods. Model of ANN with ResNet101 and XLNet achieve balanced in precision, recall and f1-score metrics for both genuine and fake content, resulting in higher F1-scores of 0.94 and accuracy of 0.90. This highlights the advantage of leveraging complementary information from text and image modalities to enhance detection accuracy.

Furthermore, the integration of features and decision fusion techniques, such as Majority Voting and Weightage Voting, enhances classification performance. Majority Voting achieves high recall of 0.89 for fake content, indicating its effectiveness in identifying fraudulent listings, albeit with a slight trade-off in precision of 0.74. This model effectively classifies fake data, achieving the highest number of correct predictions, which is 102 (shown in Fig. 8). Weightage Voting demonstrates improved precision of 0.91 for fake content as compared to previous models, suggesting its ability to reduce false positives. Furthermore, this model effectively classifies true data, achieving the highest number of correct predictions, which is 551.

Among the evaluated models, the proposed approach CWDS-DLF emerges as the top performer, achieving outstanding F1-score of 0.96 and accuracy of 0.93 metrics for both genuine and fake content. Additionally, for overall fake and genuine classifications, CWDS-DLF achieves high specificity, recall, precision, accuracy, and F1-score, as shown in

**Table 10 Performance analysis of the proposed model.** The bolded values in the table represent the best-performing results across the tested configurations.

| Unimodal (text/image/metadata) | Multimodal Feature Fusion (text and image) | Decision Fusion (text, image and metadata) | Not Fake (0) Precision | Recall | F1-score | Fake (1) Precision | Recall | F1-score | Specificity | Recall | Precision | Accuracy | F1-score | T-Test (Confidence level $p < 0.05$) |
|---|---|---|---|---|---|---|---|---|---|---|---|---|---|---|
| Resnet101 (Image) | – | – | 0.83 | **1.00** | 0.91 | **1.00** | 0.01 | 0.02 | **1.00** | 0.83 | **1.00** | 0.83 | 0.91 | 0.293 |
| 2D CNN (Image) | – | – | 0.83 | 0.99 | 0.90 | 0.25 | 0.01 | 0.02 | 0.83 | 0.25 | 0.99 | 0.83 | 0.90 | **0.007** |
| XGBoost (Metadata) | – | – | 0.92 | 0.94 | 0.93 | 0.69 | 0.63 | 0.65 | 0.92 | 0.69 | 0.94 | 0.89 | 0.93 | **0.03** |
| XLNet (Text) | – | – | 0.91 | 0.92 | 0.91 | 0.58 | 0.55 | 0.56 | 0.90 | 0.58 | 0.92 | 0.85 | 0.91 | **0.04** |
| Random Forest (Metadata) | – | – | 0.87 | 0.96 | 0.91 | 0.63 | 0.31 | 0.42 | 0.96 | 0.85 | 0.83 | 0.85 | 0.83 | **0.044** |
| FADAML (Metadata) | Concatenate the predictions from several model with the original features | – | 0.85 | 0.97 | 0.91 | 0.55 | 0.18 | 0.27 | 0.55 | 0.85 | 0.97 | 0.83 | 0.91 | **0.038** |
| – | ANN with ResNet101 and XLNet | – | 0.90 | 0.98 | 0.94 | 0.86 | 0.49 | 0.62 | 0.86 | 0.90 | 0.98 | 0.90 | 0.94 | 0.287 |
| | Spotfake+ | | 0.95 | 0.87 | 0.91 | 0.55 | 0.80 | 0.65 | 0.55 | 0.95 | 0.87 | 0.86 | 0.91 | 0.251 |
| – | MMFND | – | 0.93 | 0.93 | 0.93 | 0.67 | 0.68 | 0.67 | 0.67 | 0.93 | 0.93 | 0.89 | 0.93 | 0.055 |
| – | MMFND | Dempster–Shafer and class weightage (CWDS) | 0.94 | 0.97 | 0.95 | 0.83 | 0.70 | 0.76 | 0.83 | 0.94 | 0.97 | 0.92 | 0.95 | 0.370 |
| – | ANN with ResNet101 and XLNet | Majority Voting | **0.97** | 0.90 | 0.93 | 0.64 | **0.89** | 0.74 | 0.64 | **0.97** | 0.90 | 0.90 | 0.93 | 0.830 |
| | | Weightage Voting | 0.91 | 0.99 | 0.95 | 0.89 | 0.51 | 0.65 | 0.89 | 0.91 | 0.99 | 0.91 | 0.95 | 0.387 |
| | | Dempster–Shafer | 0.90 | 0.99 | 0.94 | 0.89 | 0.44 | 0.59 | 0.89 | 0.90 | 0.99 | 0.90 | 0.94 | 0.618 |
| | TBM | | 0.84 | 0.54 | 0.66 | 0.18 | 0.50 | 0.27 | 0.18 | 0.84 | 0.54 | 0.53 | 0.66 | **0.009** |
| | CWDS-DLF | | 0.95 | 0.97 | **0.96** | 0.82 | 0.74 | **0.78** | 0.82 | 0.95 | 0.97 | **0.93** | **0.96** | – |

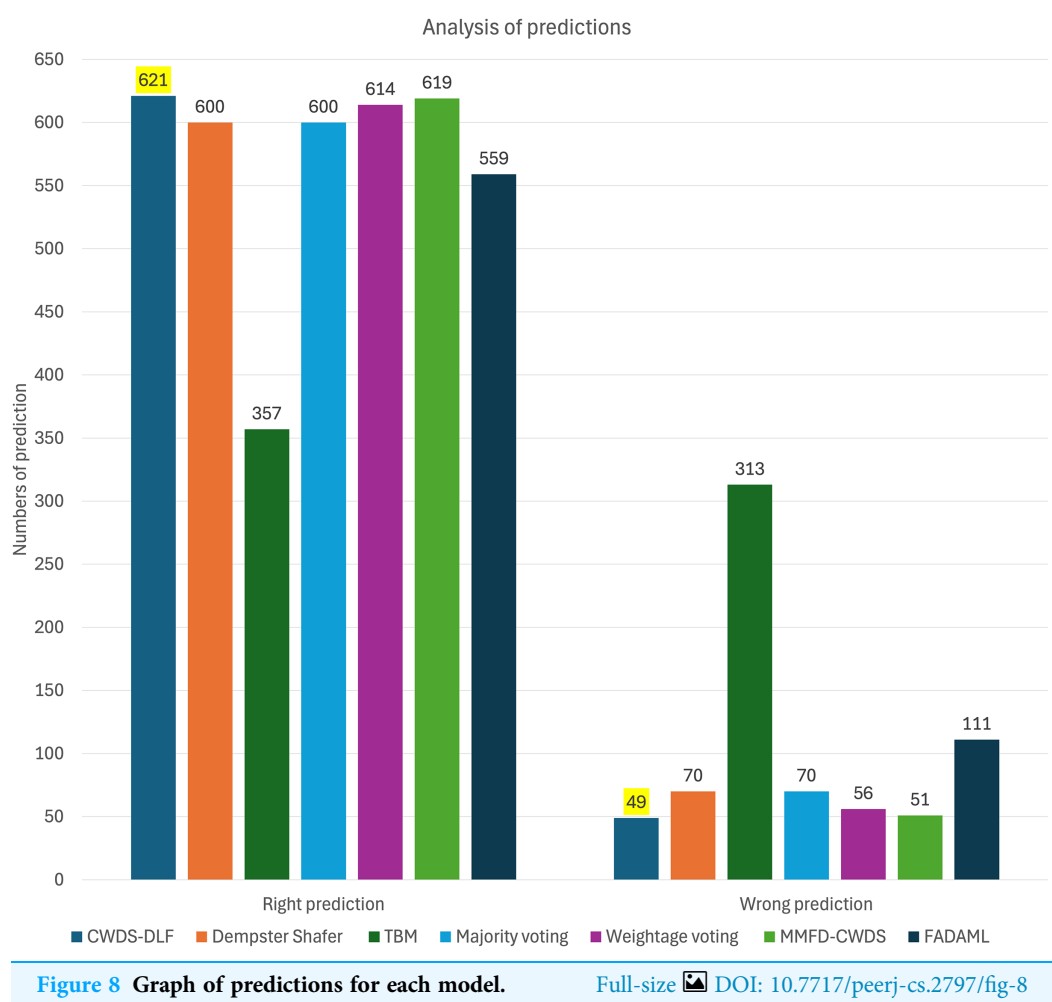

**Figure 8** **Graph of predictions for each model.**

**Table 11 Distribution of predictions for each model.** CWDS-DLF achieved the highest number of correct predictions (621) and the lowest number of incorrect predictions (49), as highlighted by the bolded values.

| Model | Right Prediction | | | Wrong Prediction | | |
|---|---|---|---|---|---|---|
| | Fake | Not Fake | Total | Fake | Not Fake | Total |
| CWDS-DLF | 85 | 536 | **621** | 30 | 19 | **49** |
| Dempster–Shafer | 51 | 549 | 600 | 64 | 6 | 70 |
| TBM | 57 | 300 | 357 | 58 | 255 | 313 |
| Majority voting | **102** | 498 | 600 | **13** | 57 | 70 |
| Weightage voting | 63 | **551** | 614 | 52 | **4** | 56 |
| FADAML | 21 | 538 | 559 | 94 | 17 | 111 |
| MMFD-CWDS | 81 | 538 | 619 | 34 | 17 | 51 |

Table 10. CWDS-DLF achieved the highest number of correct predictions (621) and the lowest number of incorrect predictions (49), as highlighted by the bolded values in Table 11 and illustrated in Fig. 8. By combining feature and decision-level fusion

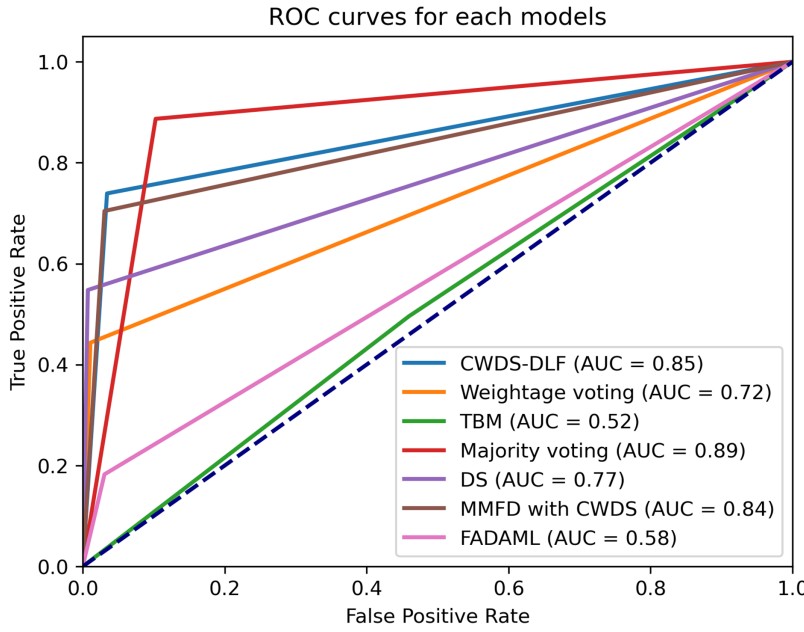

**Figure 9  ROC curves for each model.**       

techniques and leveraging class weightage, the model demonstrates superior performance in detecting fake property listings compared to other multimodal methods.

To confirm the significance of the performance improvement introduced by CWDS-DLF, a t-test was conducted. The analysis revealed that the performance of CWDS-DLF is significant at 0.05 $p$-value, affirming its effectiveness in detecting fake property listings. The results showed a significant difference between the CWDS-DLF model and several unimodal and multimodal models, including 2D-CNN at a $p$-value of 0.007, XGBoost at a $p$-value of 0.03, XLNet at a $p$-value of 0.04, MMFND at a $p$-value of 0.05, FADAML at a $p$-value of 0.038, Random Forest at a $p$-value of 0.044 and TBM at a $p$-value of 0.009.

On the other hand, some models show low values of precision, recall, and F1-score as compared to the proposed model, even if the differences are not significant. For example, the ResNet101 model (unimodal-image) has a low recall value of 0.01 and an F1-score of 0.02 as compared to the proposed model. Similarly, the ANN with ResNet101 and XLNet (multimodal-text and image) model has a recall value of 0.49 and an F1-score of 0.62, both lower than the proposed model. This statistical validation strengthens the credibility of our findings and emphasizes the importance of CWDS-DLF in addressing the challenges posed by fraudulent activities in online real estate platforms.

Furthermore, the performance of the models was compared using the AUC values *via* a graphical representation ROC. A high AUC value indicates the model's excellent ability to distinguish between positive and negative classes. As shown in Fig. 9, the Majority Voting model achieved the highest AUC value of 0.89, followed by CWDS-DLF with a value of 0.85. Although the Majority Voting model recorded the highest AUC, CWDS-DLF exhibited more balanced predictions, with the lowest false prediction rate and the highest true prediction rate compared to other models. Meanwhile, other models, such as

Weightage Voting (AUC = 0.72), DS (AUC = 0.77), and FADAML (AUC = 0.58), demonstrated lower AUC values, reflecting lower classification effectiveness. Among all models, TBM recorded the lowest performance, with an AUC of 0.52, which is only slightly better than random guessing. The diagonal dashed line in the figure, representing an AUC of 0.5, serves as the baseline for random classification. In conclusion, CWDS-DLF performed significantly better compared to other baseline models.

To further assess our model's generalization capability, we tested it on the FakeEdit dataset, which differs from our primary dataset in terms of text complexity and class distribution. The model achieved an accuracy of 0.85 and an F1-score of 0.83, maintaining performance consistency. These results indicate that the model generalizes well across different datasets, reinforcing its robustness beyond the training data.

## CONCLUSION

Our findings highlight the limitations of unimodal analyses, which often fail to accurately distinguish fraudulent listings. Models that rely solely on text, images, or metadata exhibit significant performance variations. For example, ResNet, 2DCNN and random forest models achieve high accuracy and recall in detecting non-fake content but struggle to effectively identify fake content. On the other hand, the CWDS-DLF model provides a more balanced classification performance between false and non-false classes. For instance, it achieves a precision of 0.95, recall of 0.97, and F1-score of 0.96 for non-false class data. In contrast, for false class data, it achieves a precision of 0.82, recall of 0.74, and F1-score of 0.78, which is the most optimal performance compared to other models. Overall, an accuracy of 0.93 and an F1-score of 0.96 demonstrate the highest performance compared to existing models.

Most importantly, CWDS-DLF addresses the challenge of data imbalance through the incorporation of weighting techniques. By integrating weights with the Dempster–Shafer technique, CWDS-DLF achieves superior performance compared to other conventional combined techniques such as Majority Voting, Weighted Voting, Multimodal Transformer, and Dempster–Shafer with Pignistic Probability. This is evidenced by high F1-scores and accuracy.

Furthermore, statistical analysis such as a t-test was conducted to verify the improvement in performance metrics. The results of the t-test show that the performance improvement of CWDS-DLF is statistically significant at a confidence level $p$ of 0.01 when compared to the 2DCNN, XGBoost, XLNet, MMFND, FADAML, and TBM models, further strengthening the credibility of CWDS-DLF.

Finally, the results of the analysis demonstrate that CWDS-DLF can improve user trust and security by effectively identifying fake listings on online real estate platforms. Future research efforts should explore combined techniques and extend the CWDS-DLF analysis to address other forms of fraudulent activity beyond real estate listings.

## LIMITATIONS AND FUTURE WORKS

Despite the strong performance of CWDS-DLF, several limitations must be acknowledged. First, the model's effectiveness is dataset-dependent, meaning its performance may vary

when applied to different real-world datasets. While CWDS-DLF was tested on multiple datasets, including FakeEdit, slight variations in accuracy (0.85) and F1-score (0.83) suggest that further validation on more diverse datasets is necessary to ensure robust generalization.

Second, the computational complexity of CWDS-DLF is higher than traditional unimodal models due to multimodal fusion and weighted decision-making techniques. While this enhances detection accuracy, it may also limit the model's feasibility for real-time deployment on large-scale platforms. Optimizing computational efficiency through model compression or lightweight architectures could be explored in future work. Finally, extending CWDS-DLF beyond real estate listings to detect fraudulent activity in e-commerce, job postings, or financial transactions presents a valuable research direction.

### Funding

This research was funded by Universiti Kebangsaan Malaysia (Grant code: GUP2022-060). The funders had no role in study design, data collection and analysis, decision to publish, or preparation of the manuscript.

### Grant Disclosures

The following grant information was disclosed by the authors:
Universiti Kebangsaan Malaysia: GUP2022-060.

### Competing Interests

The authors declare that they have no competing interests.

### Author Contributions

- Maifuza Mohd Amin conceived and designed the experiments, performed the experiments, analyzed the data, performed the computation work, prepared figures and/or tables, authored or reviewed drafts of the article, and approved the final draft.
- Nor Samsiah Sani conceived and designed the experiments, analyzed the data, authored or reviewed drafts of the article, and approved the final draft.
- Mohammad Faidzul Nasrudin conceived and designed the experiments, analyzed the data, authored or reviewed drafts of the article, and approved the final draft.

### Data Availability

The data is available in the Supplemental File and GitHub: https://github.com/maifuza/property-listings.

### Supplemental Information

Supplemental information for this article can be found online at http://dx.doi.org/10.7717/peerj-cs.2797#supplemental-information.

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
