# Peer review of "Class-weighted Dempster–Shafer in dual-level fusion for multimodal fake real estate listings detection"

_PeerJ Computer Science, doi:10.7717/peerj-cs.2797_

## Round 0.1 · original submission · Major Revisions

Dear Authors,

Thank you for the submission. The reviewers’ comments are now available. It is not suggested that your article be published in its current format. We do, however, advise you to revise the paper in light of the reviewers’ comments and concerns before resubmitting it. The followings should also be addressed:

1. Please pay special attention to the usage of abbreviations. Spell out the full term at its first mention, indicate its abbreviation in parenthesis and use the abbreviation from then on.
2. Equations should be used with correct equation number. Please do not use “as follows”, “given as”, etc. Explanation of the equations should also be checked. All variables should be written in italic as in the equations. Their definitions and boundaries should be defined. Provide proper reference to the governing equations.
3. Equations are part of the related sentences. Attention is needed for correct sentence formation.
4. Please recheck all the definition of variables used in the equations and further clarify these equations. All variables should be written in italic as in the equations.
5. Minor grammar and writing style errors should be corrected. Please pay special attention for correct writing, adjusting, and formatting. Especially pay special attention for the usage of space character.
6. Reviewer 1 has asked you to provide specific references. You are welcome to add them if you think they are useful and relevant. However, you are not obliged to include these citations, and if you do not, it will not affect my decision.

Warm regards,

·

Basic reporting

1. Related work:
Authors say “For example, the text structure of real estate data is complex and lengthy compared to the simple and short text found in social applications. While social application images are diverse, real estate images typically consist of houses, land, locations, and real estate environments.”
If real estate data differs significantly from social media data, then why are all the references in the related work section unrelated to real estate? Most of them focus on fake news detection and sentiment analysis, which are separate problem statements from detecting fake listings on real estate websites. The authors are requested to revise the related work/literature review to focus on studies that address the same problem statement. There has been considerable prior research in this domain; for example:

• Nguyen-Duc, Duy V. and Nguyen, Trung T. and Nguyen, Cuong V., Fake Advertisements Detection Using Automated Multimodal Learning: A Case Study for Vietnamese Real Estate Data. Available at SSRN: https://ssrn.com/abstract=4579070 or http://dx.doi.org/10.2139/ssrn.4579070

• Florea, C., Racoviţeanu, A., Florea, L., & Florea, B. (2022). Automatic Real-Estate Image Analysis for Retrieval and Classification. Bulletin of the Polytechnic Institute of Iași. Electrical Engineering, Power Engineering, Electronics Section, 68(2), 35-45.

• Javier, A.,Steven, O. “Ensuring Trust in the Real Estate Market: A Verification Program to Combat Fake Listings and Scams” – Arizona State University. https://keep.lib.asu.edu/items/192909

2. No pre-processing steps are shared for text data. How was the text data passed to the XLNet ? Stemming, lemmatization, stop word removal, icons removal, emoticons removal, special characters removal etc. are the steps required in the pre-processing of text data.

3. Figure 3 and Figure 9 are not clear, please update the figures.

4. References need to be clearer. For example, authors have quoted (Aziz, Bakar, et al., 2024). There are two similar references for this in reference section. Readers are unable to identify which one is referenced?

5. Check spelling and Grammar error. For example, author say “The existing multimodal data fusion methods have several limitations and strengths in identifying fraudulent listings.”. They should be saying ‘dilute or weakens’ in identifying fraudulent listing.

Experimental design

6. Dataset: The information on dataset is not very clearly defined. Any Deep Learning Model is as good as its data.
a) How was the data collected from the website? 12916 properties data was collected manually or some tool. Please explain.

b) “The data is categorized into two classes: fake and not fake.” – How was labelling done. How did the authors know which listing is fake or real? This is a very important step.

c) How the labeling and Metadata reviewed , what was the review process?

7. How did the authors handle labelling for multi-modal. For example, there are various scenarios in multi-modal data. This is also important. Like,
• Image is fake, text is real
• Text is fake, image is real
• Some images are real, some are fake for same property ( Each property has multiple images)
• Which image was sent to ResNet101? Cover image of each property or all images of each property. How will it work in real time?
• Image and Text are real, but metadata information is Fake.
So, when inputting the data to ResNet101 or XLNet how was the label selected on above

Validity of the findings

8. There is no comparison of the proposed work with any prior state-of-the-art methods addressing the same problem. The authors have instead compared their approach to existing frameworks of fake news detection and sentiment analysis, which were developed on social media data. However, this is not an ideal comparison, as those frameworks were specifically tuned for their respective datasets and problem statements. Given that authors acknowledge that real estate data differs significantly from social media data, they should compare their work to previous studies focused on the same problem statement.

Additional comments

9. Pease include recent research papers in the domain of multi-modal fake news. Like:
• Singh, B., Sharma, D.K. Predicting image credibility in fake news over social media using multi-modal approach. Neural Comput & Applic 34, 21503–21517 (2022). https://doi.org/10.1007/s00521-021-06086-4
• Rangel, F.; Giachanou, A.; Ghanem, BHH.; Rosso, P. (2020). Overview of the 8th Author Profiling Task at PAN 2020: Profiling Fake News Spreaders on Twitter. CEUR Workshop Proceedings. 2696:1-18. http://hdl.handle.net/10251/166528
• Singh, B., Sharma, D.K. SiteForge: Detecting and localizing forged images on microblogging platforms using deep convolutional neural network,
Computers & Industrial Engineering, Volume 162, 2021.
https://doi.org/10.1016/j.cie.2021.107733

Reviewer 2 ·

Basic reporting

The introduction is concise, with coherent arguments and motivations.
- Line 30 is incomplete (levels of...?);
- Line 55 seems to contain uninterpretable characters;
- In the 3rd paragraph, a contextualization of related works is presented, but they are not cited. I suggest inserting them;
- The reference cited in line 88 has a different font/style than the one used in the full text. The same applies to other parts of the manuscript;
- It is not clear from the reasoning in lines 106-108 whether the solution is contextual or general;
- In line 132, I suggest change "table 1" to "Table 1".

The section on related work contains a comprehensive and seemingly complete overview, considering the aim of the article.
- Since the featured text contains concepts that support the text, I suggest changing the name of the section to "Background and Related Work";
- For generic concepts (e.g., Bayesian, ensemble), it is important to provide references that support the definitions presented;
- I think it is important to provide in this section a definition of the concepts that support the study (e.g., fake real estate listings, etc.);
- Finally, it is important to point out the research gap at the end of this section. How does the study differ from previous studies?

Experimental design

The methodology is clear and well-defined, but some details are missing. The features explored, including as a baseline, are adequately described.
- Considering the feature engineering stage, the process is not clear regarding the steps taken to eliminate the 11 features that were disregarded for the study;
- According to the text, the data is categorized into two classes: "fake" and "not fake" (line 264). How were these labels constructed? Is the process reliable?
- The information in line 324 highlights that the cross-validation process was performed, but how many folds? I suggest that the process be executed more times to compute and report variation metrics (e.g., std). This would make the results presented in Table 4, for example, more solid.
- Considering unbalanced data, I suggest reporting metrics such as AUC.
- In Table 4, last column, is the reported value of macro-F1?
- It needs to be clarified how the experiments to define the parameters presented in Section 4.1 were carried out. This is clearer, for example, by considering the process of defining the hyperparameters for the XGBoost model;

Validity of the findings

The experiments carried out are satisfactory and the results are consistent. However, the discussion of the implications of the results in this section is superficial. What do the values obtained mean in practice?
- Lines 706 and 709 -- I suggest standardizing the figure to two decimal places.
- Given that only one dataset was examined, I believe the paper would benefit from a more in-depth discussion of the generalization aspects of the results.

Additional comments

Overall, in addition to appreciating the topic, I appreciated the amount of work the authors did to complete the study. The text is very well written and organized. There are still some gaps to be filled -- several of which are listed above. Finally, for the reproducibility of the study, it is important that the authors consider making codes available, etc.

---

## Round 0.2 · Minor Revisions

Dear Authors,

It seems that you have not addressed the editor's concerns and criticisms. These were following:

1. Please pay special attention to the usage of abbreviations. Spell out the full term at its first mention, indicate its abbreviation in parenthesis and use the abbreviation from then on.
2. Equations should be used with correct equation number. Please do not use “as follows”, “given as”, etc. Explanation of the equations should also be checked. All variables should be written in italic as in the equations. Their definitions and boundaries should be defined. Provide proper reference to the governing equations.
3. Equations are part of the related sentences. Attention is needed for correct sentence formation.
4. Please recheck all the definition of variables used in the equations and further clarify these equations. All variables should be written in italic as in the equations.
5. Minor grammar and writing style errors should be corrected. Please pay special attention for correct writing, adjusting, and formatting. Especially pay special attention for the usage of space character.

It is also recommended that the concerns of Reviewer 2 be addressed, and that the article be resubmitted once the necessary updates have been made.

Best wishes,

Reviewer 2 ·

Basic reporting

The changes that the authors have made to the statements in the manuscript make it easier to read. The arguments are clearer and more concise.

Minor revisions:

In the definition of "Fake real state listings" in lines 60-62, it is important to include a citation.

In line 72, "using machine learning and deep learning" seems redundant to me, as deep learning is a subset of ML.

In line 81, insert a space between the word "making" and the reference.

In line 125, it might be interesting to include references to studies conducted in the different areas of knowledge mentioned.

It might be interesting to divide Section 2 into subsections (e.g.: research gap, etc.). It is quite long.

Figure 1 is illegible.

Experimental design

The details presented/included in the methodology contributed significantly to the development of this section of the manuscript.

Review:

In step i) of preprocessing, e.g. lines 283-283, it is important to explain in general terms what these features mean, as the names are not intuitive.

For nominal attributes (Table 3), was the corresponding label assigned by the authors themselves? If so, this can be dangerous as the ML models will try to find relationships (between the numbers) that ultimately do not exist. The best option in this case is to keep categorical columns with binary information.

In Section 3.2.4, I think it is important to specify a measure of agreement between the experts (e.g.: kappa).

In Section 3.3.3. It does not indicate what percentage of data was used for training and testing. It is important to mention.

Please standardize XLNet in line 476.

Change “equations" to "Equations" in line 580.

I appreciate the fact that the authors included AUC as a metric, but for standardization reasons, they forgot to include it in Section 4.4. where the metrics explored in the paper are presented.

Validity of the findings

I still believe that the work would benefit from a more in-depth discussion of the aspects of generalization of the results. In the conclusions section, it is also important to mention the limitations of the work. In terms of the data, the technique, etc. Finally, it would be important for the scientific community at large that the authors provide the codes and data used to ensure the reproducibility of the study.

---

## Round 0.3 · accepted · Accept

Dear Authors,

Thank you for addressing the editor's and reviewers' comments. Your manuscript now seems sufficiently improved and ready for publication.

Best wishes,